# Mid-Regional Pro-Adrenomedullin Can Predict Organ Failure and Prognosis in Sepsis?

**DOI:** 10.3390/ijms242417429

**Published:** 2023-12-13

**Authors:** Silvia Spoto, Stefania Basili, Roberto Cangemi, Giorgio D’Avanzo, Domenica Marika Lupoi, Giulio Francesco Romiti, Josepmaria Argemi, José Ramón Yuste, Felipe Lucena, Luciana Locorriere, Francesco Masini, Giulia Testorio, Rodolfo Calarco, Marta Fogolari, Maria Francesconi, Giulia Battifoglia, Sebastiano Costantino, Silvia Angeletti

**Affiliations:** 1Diagnostic and Therapeutic Medicine Department, Fondazione Policlinico Universitario Campus Bio-Medico, Via Alvaro del Portillo, 200, 00128 Roma, Italy; giorgio.davanzo@unicampus.it (G.D.); domenicamarika.lupoi@unicampus.it (D.M.L.); l.locorriere@policlinicocampus.it (L.L.); f.masini@policlinicocampus.it (F.M.); g.testorio@policlinicocampus.it (G.T.); r.calarco@policlinicocampus.it (R.C.); g.battifogliaa@gmail.com (G.B.); s.costantino@policlinicocampus.it (S.C.); 2Department of Translational and Precision Medicine, Sapienza University, Viale dell’Università, 30, 00185 Rome, Italy; stefania.basili@uniroma1.it (S.B.); roberto.cangemi@uniroma1.it (R.C.); giuliofrancesco.romiti@uniroma1.it (G.F.R.); 3Departamento de Medicina Interna, Clinica Universidad de Navarra, Avda. Pío XII, 36, 31008 Pamplona, Spain; jargemi@unav.es (J.A.); flucena@unav.es (F.L.); 4Division of Infectious Diseases, Faculty of Medicine, University of Navarra, Clinica Universidad de Navarra, Avda. Pío XII, 36, 31008 Pamplona, Spain; jryuste@unav.es; 5Department of Internal Medicine, Faculty of Medicine, University of Navarra, Clinica Universidad de Navarra, Avda. Pío XII, 36, 31008 Pamplona, Spain; 6Unit of Laboratory, Fondazione Policlinico Universitario Campus Bio-Medico, Via Alvaro del Portillo, 200, 00128 Roma, Italy; m.fogolari@policlinicocampus.it (M.F.); m.francesconi@policlinicocampus.it (M.F.); s.angeletti@policlinicocampus.it (S.A.); 7Research Unit of Clinical Laboratory Science, Department of Medicine and Surgery, Università Campus Bio-Medico di Roma, Via Alvaro del Portillo, 21, 00128 Roma, Italy

**Keywords:** sepsis, septic shock, organ failure, mid-regional pro-adrenomedullin (MR-proADM), acute kidney injury (AKI), acute respiratory distress syndrome (ARDS), Glascow Coma Scale (GCS), acute heart failure (AHF), Sequential Organ Failure Assessment (SOFA), intensive care unit (ICU)

## Abstract

Sepsis causes immune dysregulation and endotheliitis, with an increase in mid-regional pro-adrenomedullin (MR-proADM). The aim of the study is to determine an MR-proADM value that, in addition to clinical diagnosis, can identify patients with localized infection or those with sepsis/septic shock, with specific organ damage or with the need for intensive care unit (ICU) transfer and prognosis. The secondary aim is to correlate the MR-proADM value with the length of stay (LOS). In total, 301 subjects with sepsis (124/301 with septic shock) and 126 with localized infection were retrospectively included. In sepsis, MR-proADM ≥ 3.39 ng/mL identified acute kidney injury (AKI); ≥2.99 ng/mL acute respiratory distress syndrome (ARDS); ≥2.28 ng/mL acute heart failure (AHF); ≥2.55 ng/mL Glascow Coma Scale (GCS) < 15; ≥3.38 multi-organ involvement; ≥3.33 need for ICU transfer; ≥2.0 Sequential Organ Failure Assessment (SOFA) score ≥ 2; and ≥3.15 ng/mL non-survivors. The multivariate analysis showed that MR-proADM ≥ 2 ng/mL correlates with AKI, anemia and SOFA score ≥ 2, and MR-proADM ≥ 3 ng/mL correlates with AKI, GCS < 15 and SOFA score ≥ 2. A correlation between mortality and AKI, GCS < 15, ICU transfer and cathecolamine administration was found. In localized infection, MR-proADM at admission ≥ 1.44 ng/mL identified patients with AKI; ≥1.0 ng/mL with AHF; and ≥1.44 ng/mL with anemia and SOFA score ≥ 2. In the multivariate analysis, MR-proADM ≥ 1.44 ng/mL correlated with AKI, anemia, SOFA score ≥ 2 and AHF. MR-proADM is a marker of oxidative stress due to an infection, reflecting severity proportionally to organ damage.

## 1. Introduction

It has been found that sepsis leads to inflammation. This occurs through the release of C-reactive protein and neopterin and oxidative stress, with the increased production of reactive oxygen species (ROS) and reactive nitrogen species (RNS) and a rise in nitric oxide levels. This involves cell damage by lipid peroxidation, nucleic acid damage and endothelial dysfunction, leading to organ failure [1,2].

Mortality due to sepsis is caused by several factors, including the diagnostic and therapeutic timing from disease onset, the immunoregulatory phenotype of the host, the bacterial load and the multidrug-resistant phenotype of the pathogen [3].

The degree of inflammatory response dysfunction and the level of organ dysfunction in response to infection correlate with the prognosis of sepsis [1].

The sepsis poor prognosis phenotype is characterized by several factors: enhanced spread from the site of infection due to decreased local defenses, a higher bacterial load, increased aggressiveness of the infection, the presence of multidrug-resistant bacteria in the bloodstream, endothelium damage and a dysregulated proinflammatory response.

Some studies have suggested that sepsis onset could lead to immune system cell damage and endotheliitis, resulting in biohumoral dysregulation, inflammation with oxidative stress and the increased expression of adrenomedullin (ADM) [1].

The increase in ADM levels leads to hypotension, hypoxia, vascular leakage, inflammation and an increased state of infection with myocardial injury, immunothrombosis and subsequently hypoperfusion, with the establishment of other multiple organ dysfunctions—acute kidney or liver damage, acute respiratory distress syndrome (ARDS) and pulmonary embolism, ischemic stroke and intestinal and other systems’ damage [1].

Myocardial dysfunction due to ADM expression could be a cornerstone of systemic hypotension and organ hypoperfusion, resulting in a reduction in cardiac output with overload and systemic and pulmonary edema.

ADM is a peptide of 52 amino acids, a member of the CT/CGRP superfamily whose gene is located on the 11 chromosome. It expresses the onset of organ damage, its severity, the degree of edema and prognosis [4,5,6,7,8,9,10,11,12,13,14].

The precursor peptide, mid-regional pro-adrenomedullin (MR-proADM), is more stable than ADM, released in 1:1 ratio to ADM. MR-proADM shows high predictiveness regarding prognosis during severe inflammation, such as in the case of sepsis, acute heart failure (AHF) or COVID-19 [8,9,10,11,15,16,17,18,19,20,21,22,23,24,25,26,27,28,29].

MR-proADM has proven to be superior to NT-proBNP (the “gold standard” biomarker) in the case of tissue congestion, such as edema and lung congestion [12,13]. Furthermore, MR-proADM significantly correlates with myocardial injury, identifying patients at high risk of death who could benefit from adrecizumab therapy [5,10,14,30,31,32].

The correlation between high bio-ADM values and organ dysfunction (with high Sequential Organ Failure Assessment (SOFA) score and cardiovascular SOFA subscore; the requirement for vasopressors/inotropes, high fluid volume resuscitation or renal replacement therapy; long intensive care unit (ICU) stays; and mortality) and the benefit from adrecizumab administration has recently been demonstrated in septic patients admitted to the ICU [33,34,35,36,37].

Furthermore, the persistence of high bio-ADM values at day 2 after admission to intensive care was associated with prolonged organ dysfunction and high mortality [34,35].

Although ADM is ubiquitous, the prevalent expression of the peptide and receptor representation is at the level of the adrenal medulla, heart (cardiac atria), lungs, kidney and central nervous system, which can lead to the commencement of organ damage [38,39,40,41,42,43,44].

The quantity of stimuli that cause ADM overexpression depend on the length of time for which the stimulus lasts and the immunoregulatory status of the host, which depends on the genetic inheritance of the immune response, including the ADM expression capacity and prevalence, and the distribution of ADM receptors may determine the prognosis of sepsis.

MR-proADM has been proposed as a useful early diagnostic biomarker of serious infection in critically ill patients, even in post-surgical states, and its concentrations correspond to microcirculatory and endothelial damage in the early stages of organ dysfunction, before the development of organ failure and therefore also in patients with a low SOFA score [17,35,36,37,38].

It is important to note that the diagnosis of suspected sepsis is clinical and that the MR-proADM value is a biomarker that must be added to the clinical diagnosis and other biohumoral tests.

In the literature, there is no consensus on a value of MR-proADM that is diagnostic or prognostic or expresses specific organ damage, its quantification or the need for ICU transfer [7,16,45,46,47,48,49].

The aim of the present study is to define an MR-proADM value that, in addition to clinical diagnosis, can identify patients with localized infection or those with sepsis/septic shock, with specific organ damage or with the need for ICU transfer and prognosis.

A secondary aim is to correlate the MR-proADM value with the length of stay (LOS).

The added value of the study is to provide the clinician with an MR-proADM value that, in addition to the clinical diagnosis, can identify patients (a) with localized infection or sepsis/septic shock, (b) with specific organ damage, (c) with the need for ICU transfer and (d) with a prognosis, to be able to treat them as appropriately, promptly and intensively as possible, saving lives.

## 2. Results

### Baseline Characteristics of Included Population

The study population was composed of 301 patients affected by septic syndrome, subclassified into 177 patients with sepsis and 124 patients with septic shock, and of 126 patients with localized infection without sepsis. In Table 1, the demographic and laboratory characteristics and clinical scores of the population under study are presented.

In septic patients, the median age was 74.5 [IQR 66.0–82.0], and in septic shock patients, it was 73 years [IQR 66.0–82.0] and 77 years [IQR 68.0–83.0], respectively, being higher in septic shock patients (*p* = 0.019). Comparing the median age between the septic and localized infection groups, the median age was significantly higher (*p* = 0.0003) in the latter group (79 years [69.8–82.0]) (Table 1). The median age of patients with localized infection was comparable with the median age of septic shock patients (*p* = 0.08).

Regarding sex, the male prevalence was comparable between the three groups of patients.

As regards comorbidities, cardiovascular diseases, diabetes, chronic obstructive pulmonary disease (COPD) and chronic kidney diseases were the most prevalent, followed by cerebrovascular disease, cancer and liver diseases (Table 1).

In particular, cancer prevalence was comparable between sepsis and septic shock patients, while it was more prevalent in the localized infection than the septic group (*p* = 0.0001); COPD prevalence was equally represented within the septic group and more prevalent in patients with localized infection (*p* = 0.002); cardiovascular disease was more represented in patients with septic shock than sepsis (*p* = 0.028) and more prevalent in patients with localized infection than sepsis (*p* = 0.0017); liver disease prevalence was comparable in all three groups of the study population; chronic kidney disease was equally represented within the septic group, while its prevalence was higher in the case of localized infection than sepsis (*p* = 0.036); diabetes was equally distributed in the study population; cerebrovascular disease prevalence was comparable in sepsis and septic shock patients, whereas it was more represented in patients with localized infection than sepsis (*p* < 0.0001). The results are reported in Table 1.

In the sepsis population 252 patients/301 (83.7%) presented SOFA scores ≥ 2 and 21/301 (7%) had q-SOFA scores ≥ 2. The remaining 28 septic patients/301 (9.3%) had microbiological isolation on blood, procalcitonin elevation and at least one SOFA and one q-SOFA criterion. In patients with localized infection, 71/126 (56%) had SOFA scores ≥ 2 and 1/126 (0.8%) had a q-SOFA score ≥ 2. The presence of a SOFA score ≥ 2 was significantly higher (*p* < 0.0001) in the septic group than the localized infection group, and in the case of septic shock patients than sepsis patients (*p* = 0.003). The same results were observed for a q-SOFA score ≥ 2 (septic group vs. localized infection, *p* = 0.0084; septic shock vs. sepsis *p* = 0.027).

Among the population of sepsis patients, organ damage was present in 263/301 (87.3%), and 185/301 patients (61.5%) were affected by ≥ 2 organs’ failure. The prevalence was significantly higher (*p* < 0.0001) in the case of septic shock (exactly 86/177 (48.6%) patients with sepsis and 99/124 (80%) with septic shock). Single organ involvement was present in 78/301 (26%) patients within the septic group, 63/177 (35.6%) with sepsis and 15/124 (12%), showing a higher prevalence (*p* < 0.0001).

In patients with localized infection, organ damage was present in 101/126 (80%); the prevalence was comparable to that of the septic group (*p* = 0.06). Single organ involvement was diagnosed in 41/126 (32.5%), while multiple organ involvement was seen in 60/126 (47.6%), being significantly lower in the septic group (*p* < 0.0001).

The stratification of organ damage in septic patients and those with localized infection is presented in Figure 1.

As for organ damage in septic patients, the most affected areas of the body appeared to be the cerebral (52.2%) and kidney (53.7%) regions, followed by ARDS, cardiac injury and liver impairment (42.5%, 33.2% and 16.6%, respectively). Conversely, in patients with localized infection, the most affected was the heart (44%), followed by ARDS (31.7%), AKI (23%) and liver impairment (4.7%); none presented GCS < 15.

Anemia was detected in 270/301 (89.75) patients in the study population, being more prevalent (*p* = 0.027) in septic shock (117/124 (94.3%)) than sepsis patients (153/177 (86%)). Moreover, 76 out of 126 patients with localized infection had anemia (60.3%), and the prevalence was significantly higher in septic patients (*p* < 0.0001).

Acute liver failure was detected in 50/301(16.6%) subjects with sepsis, of which 26/124 (21%) had septic shock and 24/177 (13.6%) had sepsis, showing a comparable prevalence between the two groups. In infection patients, it was diagnosed in 6/126 (4.7%), a significantly lower proportion than in the sepsis group (*p* = 0.0009).

AKI was present in 157/301 (52%) of patients with sepsis, 77/124 (62%) with septic shock and 80/177 (45%) with sepsis, with a significantly higher prevalence in the first group (*p* = 0.0037). In the case of localized infection, AKI was detected in 29/126 (23%), which was significantly lower compared to patients with sepsis or septic shock (*p* < 0.0001).

AKI classified according to the RIFLE criteria has been shown to have a greater influence than ARDS on prognosis.

Specifically, of the 301 septic patients, 157 (52.2%) developed AKI and, of these, 63 (40.1%) were type R, 50 (31.8%) type I, 31 (19.7%) type F and 10 (6.4%) type L, and three (1.9%) required renal replacement therapy (group E).

Of the 301 septic patients, 128 (42.5%) developed ARDS, of which 66 (51.6%) were type I, 39 (30.5%) type II and 6 (4.7%) type III.

The onset of AKI, an impaired mental state, septic shock and the presence of multi-organ dysfunction (with MR-proADM cut-off ≥ 3.38) were the impairments most affecting prognosis.

Mortality at 30 days was significantly correlated with AKI, an impaired mental state and shock; indeed, the risk of mortality increased 2.5-fold if there was AKI and 5-fold if there was GCS < 15 and shock.

AHF was present in 101/301 (33.5%) patients with sepsis, of which 57/124 (46%) had septic shock and 44/177 (23%) had sepsis, showing a significant difference (*p* < 0.0001) between the two groups. Moreover, 56 out of 126 patients with infection had AHF (44.4%), and the prevalence was significantly higher than in the sepsis group (*p* = 0.03). The need for catecholamine administration correlated significantly with death at 30 days and multi-organ damage (odds ratio = 6.48 and 9.87, respectively), whereas an MR-proADM value ≥ 2.35 at admission discriminated those with anemia, but this did not correlate with mortality or the presence of organ dysfunction.

In 290/301 (96%) subjects with sepsis, GCS was <15 but the prevalence was significantly higher in the case of septic shock (*p* = 0.0003), specifically in 80/124 (64.5%) patients, vs. 77/177 (43.5%) for sepsis. All patients with localized infection had GCS > 15.

Organ failure involving one organ was diagnosed in 78/301 (26%) septic patients, while organ failure of two or more organs was detected in 185/301 (61.5%). Single organ involvement was predominantly found in patients without septic shock (63/177, 35.6% vs. 15/124, 12% septic shock). Conversely, the extensive involvement of two or more organs was significantly more prevalent (*p* < 0.0001) in septic shock patients, at 99/124 (79.8%), vs. sepsis in 86/177 (48.6%). In patients with localized infection, single organ failure was detected in 41/126 (32.5%) and multiple organ failure in 60/126 (47.6%). Multiple organ involvement was significantly more prevalent in the group of patients with sepsis or septic shock (*p* = 0.0001), whereas single organ involvement was comparable between the two groups.

The need for ICU transfer was noted in 66/301 (21.9%) patients in the septic population; it was significantly higher (*p* = 0.0014) in the case of septic shock, as exactly 40/124 (32.2%) septic shock patients needed ICU assistance, vs. 29/177 (16.4%) septic patients and 0% in the population with localized infection. ARDS was present in 128/301 (42.5%) of patients included in the sepsis and septic shock population; it was significantly higher (*p* < 0.0001) in the case of septic shock, affecting 70/124 (56.4%) patients, versus 58/177 (32.7%) of those with sepsis. Forty of the 126 patients with localized infection presented ARDS (31.7%), and the prevalence was significantly higher in the case of sepsis (*p* = 0.037).

The need for catecholamine administration was registered in 88/301 patients; it was significantly higher (*p* < 0.0001) in those with septic shock at 83/124 (67%), vs. 5/177 (2.8%) of those with sepsis. No patient with localized infection received catecholamine administration.

In the septic population, the median LOS was 15 [IQR −10.5–26.5] days. It was comparable between septic shock patients, at 18 days [IQR −8.2–36.2], and sepsis patients, at 15 days [IQR −11–26]. In patients with localized infection, the median LOS was 10 days [IQR 8–13 days]. The LOS was significantly longer in the sepsis group (*p* < 0.0001).

In regard to prognosis, the mortality rate in the sepsis and septic shock population was 26.2% (79 out of 301). Dividing the population between sepsis and septic shock, 52/124 (42%) patients with septic shock died, vs. 27/177 (15.2%) patients with sepsis. The mortality rate was significantly higher in the case of septic shock (*p* < 0.0001), whereas, in the group with localized infection, the mortality rate was 0%.

In sepsis and septic shock patients, the median MR-proADM was 2.5 ng/mL (IQR: 1.6–4.3 ng/mL). The median value was significantly higher in patients with septic shock (*p* < 0.0001). Specifically, it was 3.65 ng/mL (interquartile range 25–75th percentile: 2.0–6.4 ng/mL) in septic shock vs. 2.1 ng/mL (interquartile range 25–75th percentile: 1.4–3.2 ng/mL).

In patients with localized infection, the median MR-proADM was 1.19 ng/mL (interquartile range 25–75th percentile: 0.84–1.87 ng/mL). The median value was significantly higher in patients with sepsis and septic shock (*p* < 0.0001).

The median values of creatinine, bilirubin, INR, AST, ALT and INR were similar between the two sub-groups of sepsis and septic shock patients, while the median value of lactate was significantly higher in the septic shock sub-group (*p* < 0.0001). Specifically, it was 18.4 mmol/L (interquartile range 25–75th percentile: 13.0–28 mmol/L) in septic shock vs. 12 mmol/L (interquartile range 25–75th percentile: 9.5–14.5 mmol/L).

ROC curve analysis showed that MR-proADM values at admission ≥ 3.39 ng/mL significantly identified septic patients with AKI (AUC = 0.684, *p* < 0.001) (Figure 2A); ≥2.99 ng/mL with ARDS (AUC = 0.614, *p* = 0.001); ≥2.28 ng/mL with AHF (AUC = 0.584, *p* = 0.007); ≥2.55 ng/mL with GCS < 15 (AUC = 0.632, *p* < 0.001); two or more organs’ involvement ≥ 3.38 (AUC = 0.689, *p* < 0.001) (Figure 2B); and the need for ICU transfer ≥ 3.33 (AUC= 0.643, *p* < 0.001) (Figure 3A). MR-proADM identified patients with SOFA scores ≥ 2 at cut-off ≥ 2.0 ng/mL (AUC = 0.77, *p* < 0.001), whereas it did not identify patients with positive q-SOFA scores or those with acute liver failure.

In the ROC curve analysis, MR-proADM at admission ≥ 3.15 ng/mL (AUC = 0.702, *p* < 0.001) significantly discriminated non-survivors from survivors (Figure 3A). Specifically, at cut-off ≥ 3.39 ng/mL (AUC = 0.76, *p* < 0.001) (Figure 3B), it identified patients who died at 30 days; at cut-off ≥ 4.0 ng/mL (AUC = 0.77, *p* < 0.001), it identified patients who died at 90 days (Figure 3C); and at cut-off ≥ 3.33 (AUC = 0.64, *p* < 0.001), it identified patients who needed an ICU transfer (Figure 3C).

Univariate and multivariate analyses were performed to evaluate the correlation of MR-proADM values at admission above the cut-off of 2 ng/mL and 3 ng/mL as well. Results are shown in Table 2 and Table 3.

In the univariate analysis, a significant correlation was found between MR-proADM ≥ 2 ng/mL and AKI, anemia, ARDS, a need for ICU transfer, a need for catecholamine administration, shock, a SOFA score ≥ 2, AHF, GCS < 15 and multiple organ involvement (Table 2), and between MR-proADM ≥ 3 ng/mL and AKI, anemia, ARDS, a need for catecholamine administration, shock, a SOFA score ≥ 2, GCS < 15 and multiple organ involvement (Table 2). Significant odds ratios of risk were calculated and are reported in Table 2.

In the multivariate logistic regression analysis, a significant correlation was found between MR-proADM ≥ 2 ng/mL and AKI, anemia and a SOFA score ≥ 2, and between MR-proADM ≥ 3 ng/mL and AKI, GCS < 15 and a SOFA score ≥ 2 (Table 3).

With regard to the correlation with prognosis, the univariate analysis showed that 30-day mortality and 90-day mortality significantly correlated with AKI, ARDS, GCS < 15, a need for ICU transfer, a need for catecholamine administration, multiple organ failure, MR-proADM ≥ 2 ng/mL at admission, MR-proADM ≥ 3 ng/mL at admission, a SOFA score ≥ 2 and shock, for which the corresponding significant odds ratios are reported in Table 4. Conversely, 30-day and 90-day mortality did not correlate with acute liver and heart failure, q-SOFA scores ≥ 2 or anemia (Table 4).

In the multivariate logistic regression analysis, there was a significant correlation between 30-day mortality and AKI, GCS < 15, a need for ICU transfer and a need for catecholamine administration, with significant odds ratios as reported in Table 5. Meanwhile, the multivariate logistic regression analysis showed a significant correlation between 90-day mortality and AKI, GCS < 15 and a need for ICU transfer, with significant odds ratios as reported in Table 5.

In the multivariate logistic regression analysis, including in the model stratification of patients with AKI by the RIFLE criteria and ARDS severity (grade 2 and 3), variables influencing mortality at 30 days were AKI with RIFLE class E and F, GCS < 15, multiple organ involvement, the need for ICU transfer and catecholamine administration (Table 6). Meanwhile, in the multivariate logistic regression analysis, significant variables related to 90-day mortality were AKI, GCS < 15, ICU transfer and SOFA score ≥ 2 (Table 6).

In our studied septic populations, the most represented forms of organ damage were an impaired state of consciousness (290/301, 96%) and AKI (152/301, 52.2%), followed by ARDS and AHF (128/30, 42.5% and 100/301, 33.2%, respectively).

Our study identified specific MR-proADM cut-offs for organ damage, with specific values ≥ 2.28 nmol/L in the case of AHF, ≥2.99 in ARDS, ≥3.39 in the presence of AKI, ≥2.55 in the case of an impaired mental state, ≥3.38 in the case of increased disease severity defined by multiple organ involvement (such as in the case of ≥2 organ damage), ≥3.36 in the case of septic shock, ≥3 in the case of death (specifically, ≥3 in the case of 90-day mortality or ≥4 in the case of 30-day mortality) and ≥3.33 when an ICU transfer was necessary.

Furthermore, a MR-proADM value ≥ 2 identified patients with a SOFA score ≥ 2 (AUC = 0.77 *p* < 0.001), while it did not identify patients with q-SOFA ≥ 2.

In the 126 patients with localized infection, the MR-proADM median value was 1.19 ng/mL [IQR 0.84–1.87 ng/mL]. In these patients, the ROC curve analysis showed that MR-proADM values at admission ≥ 1.44 ng/mL significantly identified patients with AKI (AUC = 0.823, *p* < 0.001); ≥1.0 ng/mL significantly identified patients with AHF (AUC = 0.749, *p* < 0.0001); and ≥1.44 ng/mL significantly identified patients with anemia (AUC = 0.699, *p* < 0.001) and with a SOFA score ≥ 2 (AUC = 0.679 *p* < 0.001). Meanwhile, it did not discriminate patients with positive q-SOFA scores from those with acute liver failure or ARDS.

In the univariate analysis and multivariate logistic regression, a significant correlation was found between MR-proADM ≥ 1.44 ng/mL and AKI, anemia, a SOFA score ≥ 2 and AHF (Table 7 and Table 8).

## 3. Discussion

### 3.1. MR-proADM Levels and Organ Damage and Outcome

Data from our study show that the prevalence of organ damage is independent of whether or not myocardial damage is evidenced by AHF and/or atrial fibrillation with AHF. Thus, endotheliitis is the true cornerstone of the onset of organ damage and prognosis, rather than myocardial damage. Sepsis determines inflammation, oxidative stress and endotheliitis, which is expressed as damage to the endothelial barrier, a reduction in antimicrobial properties, a reduction in the vasodilating effect leading to edema and hemodynamic overload, leading to organ dysfunction. Recently, it has been demonstrated in vivo and in vitro how the endogenous peptide adrenomedullin serum level increases in production and expression during severe infection, because it is an efficient counter-regulatory molecule with the purpose of the regulation and stabilization of the endothelial barrier and the protection of the microcirculation and then of the hemodynamic balance [50,51,52,53].

Furthermore, the study shows that MR-proADM is a good indicator of organ damage both in patients with localized infection and in patients with sepsis and/or septic shock, at different cut-off values.

The organ damage that is shown to be most relevant and thus correlates with higher MR-proADM values is AKI, both in infected and septic patients. In septic patients, as well as AKI, brain damage is also relevant.

In the univariate analysis, in septic patients, a MR-proADM cut-off ≥ 2 or ≥3 ng/mL correlates significantly with AKI, anemia, AHF, GCS < 15, a need for ICU transfer, a need for catecholamine administration, SOFA score ≥ 2, AHF, multiple organ damage and shock, but not with acute liver failure or q-SOFA ≥ 2.

In the multivariate analysis, the correlation remained significant only for AKI, GCS < 15, the SOFA score and the need for ICU transfer. This shows that MR-proADM identifies patients who have one or more organ failures, even if the correlation is stronger for some variables (AKI, GCS < 15, need for ICU transfer) that reflect more severe phenotypes.

The multivariate logistic regression analysis, including in the model of AKI classified by the RIFLE criteria and ARDS by severity grade (grade 2 and 3), showed that mortality at 30 days was correlated with AKI, RIFLE classes E and F, GCS < 15, multiple organ involvement, the need for ICU transfer and catecholamine administration (Table 6), whereas 90-day mortality correlated significantly with AKI of any RIFLE class, GCS < 15, ICU transfer and a SOFA score ≥ 2 (Table 6). The results from our study showed that short-term prognosis, as mortality at 30 days from sepsis onset, was significantly influenced by severe AKI (RIFLE E and F), an impaired mental status (GCS < 15) and multi-organ involvement. This is affirmed by the association with the need for ICU transfer and a SOFA score ≥ 2. The same trend was observed for 90-day mortality, with the exception of the AKI influence, as any class of RIFLE was associated, suggesting that AKI at any stage, including a less severe AKI condition, could be determinant for long-term prognosis in terms of 90-day mortality.

Having shown that MR-proADM correlated with organ damage, we also tested its correlation with prognosis, which was significant. This reinforces data already described in the literature [6,7,17,18,19,20,21,22,23,49,54,55,56,57,58,59,60,61,62].

In our study, a cohort of 126 patients with localized infection without sepsis was included. In these patients, an MR-proADM value at admission ≥ 1.44 ng/mL identified the potential presence of AKI in the presence of an infectious state. This result suggests that among possible organ failures, AKI is involved even during localized infection, as confirmed in the univariate and multivariate logistic regression analyses. Besides AKI, in these patients, an MR-proADM value above the cut-off value is associated with anemia and AHF. Moreover, in these patients, the presence of organ damage is confirmed by a SOFA score ≥ 2, which is significantly associated.

The value of MR-proADM is an indicator that is directly proportional to the severity of the organ damage and to the prognosis. MR-proADM is a marker of sepsis that reflects the level of oxidative stress and, thus, the severity of the disease proportionally to organ damage and prognosis.

The use of biomarkers assists in the clinical diagnosis of infection. Clinical sepsis scores allow septic patients to be stratified based on the number of organs damaged and the severity of organ damage; therefore, they become positive later than the elevation of MR-proADM, which, moreover, can be affected by oxidative stress due to other, even non-infectious causes.

In our study, the established cut-off value of MR-proADM (corresponding to the best sensitivity and specificity values) to identify patients with localized infection was ≥1.44 ng/mL; that to identify patients with sepsis with a SOFA score ≥ 2 or to identify those with a need for ICU transfer or not surviving at 90 days was ≥2 ng/mL.

The use of these cut-offs could allow timely and intensive treatment, avoiding the onset of sepsis and organ damage and/or death. These results are in line with a recent meta-analysis and systematic review that evaluated the diagnostic value of MR-proADM in sepsis, finding that MR-proADM is an excellent biomarker for the diagnosis of sepsis [7,16,46,47,48,49].

In the sepsis population, using 1–1.5 nmol/L as the cut-off value of MR-proADM led to higher combined sensitivity and specificity for the diagnosis of sepsis, with values of 0.83 and 0.90, respectively [16].

Noteworthy is the essential aid provided by the use of a cut-off of MR-proADM ≥ 1.5 nmol/L for early sepsis diagnosis in those with a negative SOFA score. In this study, indeed, approximately 35% of patients were negative for SIRS criteria or q-SOFA and SOFA scores or for all of them, despite evidence of a positive blood culture and documented microbiological isolate or clinical diagnosis of infection. In these patients, the use of MR-proADM was crucial to provide early diagnosis and confirm the suspicion of sepsis [7,47].

These MR-proADM cut-offs also correspond to those found in another recent work by S. Graziadio, in which an MR-proADM value greater than 1.5 nmol/L correlated with an acuity increase, while a value greater than 1.89 nmol/L correlated with a deterioration in patients admitted to hospital with a mild to moderately severe acute illness corresponding to a National Early Warning Score (NEWS) between 2 and 5 [62]. 

MR-proADM had high accuracy in identifying both 28-day and 90-day mortality, compared to all other biomarkers and clinical scores [46,49].

Outside the ICU, an MR-proADM cut-off value > 3.39 nmol/L in sepsis and > 4.33 nmol/L in septic shock was associated with a significantly higher risk of 90-day mortality [19].

In ICU patients admitted with SIRS and organ dysfunction, an MR-proADM cut-off point of 1.425 nmol/L helped to identify those with sepsis, while an MR-proADM value above 5.626 nmol/L, 48 h after admission was associated with a high risk of death [49].

The added value of our study is the effort to establish a threshold in the evaluation of MR-proADM that may allow for the different management of patients with sepsis. In clinical use, the MR-proADM value distinguishes septic patients at a higher risk of death by identifying those who may also benefit from more strict medical treatment, including hemodynamic management, infection source control, intensive and timely antibiotic therapy and the modulation of host response therapy also with adrecizumab, with administration as early as possible upon evidence of sepsis with AKI, impaired GCS or shock [63,64,65,66].

Our study, therefore, identifies the phenotype of sepsis and infection patients with the worst prognosis and also considers compliance with the “antimicrobial stewardship” rules to identify patients who need targeted or early empirical antibiotic therapy based on the severity characteristics of each patient [37,50,51,63,65,67,68].

### 3.2. Limitations and Perspectives

The main limitation of our study is the absence of an external validation cohort. To ascertain the generalizability of the cut-off that we identified within our population, future investigations will be necessary, encompassing diverse centers and settings. However, many pathophysiological processes in sepsis are still to be investigated.

Furthermore, the control group was chosen as real-life patients matched for age and sex as much as possible. Although this population consisted of older patients, who had a higher frequency of cancer, chronic kidney disease, cerebrovascular disease and diabetes, an age comparison in patients with septic shock and a control group was not significant (*p* = 0.89).

The challenge, which may represent a limitation in this monocentric study, is that of the difficulty of hypothesizing and deducing from the evidence of the synthesis of clinical and biohumoral data a pathophysiological process that is still under study. However, we hope that these new data represent a step forward in the exploration of such an important and health-critical topic.

## 4. Materials and Methods

### 4.1. Patient Selection and Study Design

A retrospective study was performed on 301 randomly selected consecutive patients with sepsis or septic shock and 126 patients with localized infection admitted to the Diagnostic and Therapeutic Medicine Department and General Surgery Department of the Fondazione Policlinico Universitario Campus Bio-Medico of Rome, between May 2018 and June 2023.

Informed consent was obtained from all patients at hospital admission.

### 4.2. Ethical Statement

The study was conducted according to the guidelines of the Declaration of Helsinki and approved by the Ethical Committee of the Fondazione Policlinico Universitario Campus Bio-Medico of Rome (28.16 TS Com Et CBM).

Inclusion criteria were as follows: patients affected by sepsis or septic shock and with localized infection. The diagnosis of sepsis was defined by the Third Consensus Sepsis Conference and compared with PCT and MR-proADM [36,37].

Exclusion criteria were the absence of informed consent and pregnancy.

At admission (T 0), demographic characteristics were recorded, such as age, gender, immune status (active malignancy or other causes of an immunocompromised state), comorbidities and clinical presentation. A physical examination including a cardiac, abdominal, respiratory and neurological evaluation was performed for each patient.

### 4.3. Clinical Scores, Laboratory Parameters and Blood Gas Analysis

All patients received a complete physical examination, including body temperature; blood pressure; heart and respiratory rate; cardiac, pulmonary, abdominal and neurological evaluation; hemogasanalysis; electrocardiogram; transthoracic echocardiography; and imaging if clinically needed.

The following clinical and laboratory parameters were registered at inclusion (day 0): Glascow Coma Scale (GCS); acute respiratory distress syndrome (ARDS); acute kidney injury classified by Risk, Injury, Failure, Loss of Kidney Function and End-Stage Kidney Disease (RIFLE) criteria classification; quick Sequential Organ Failure Assessment (q-SOFA); Sequential Organ Failure Assessment (SOFA); atrial fibrillation; the use of inotropes/catecholamines; hemoglobin (Hb); platelets (PLT); alanine aminotransferase (ALT) and aspartate aminotransferase (AST); international normalized ratio (INR); bilirubin; creatinine; procalcitonin (PCT); MR-proADM; lactate; and PaO_2_/FIO_2_. These were performed at diagnosis and when clinically necessary.

### 4.4. Definitions and Laboratory Parameters

The diagnosis of sepsis and septic shock was performed according to the Third Consensus Conference Criteria when the q-SOFA or SOFA score was ≥2 from baseline, in the presence of an infection [36,37].

The diagnosis and treatment of sepsis, pneumonia, urinary tract, intra-abdominal, skin, soft tissue, bloodstream and all other included infections were managed according to the currently available international guidelines [36,37,69,70,71,72,73].

The diagnoses and clinical scoring of GCS, ARDS and AHF RIFLE criteria classification were defined according to the most up-to-date international guidelines [63,64,65,66].

Patients defined as having septic shock requiring mechanical ventilation and/or plasma ultrafiltration were transferred to the ICU.

Anemia was defined as hemoglobin (Hb) levels < 12.0 g/dL or < 13.0 g/dL in women or in men, respectively, according to the World Health Organization (WHO) [67].

### 4.5. MR-proADM Plasma Measurement

MR-proADM plasma concentrations were measured by an automated Kryptor analyzer, using a time-resolved amplified cryptate emission (TRACE) technology assay (Kryptor PCT; Brahms AG, Hennigsdorf, Germany), with commercially available immunoluminometric assays (Brahms) [7,17]. MR-proADM measurement was performed only at admission (T = 0), since the marker has slow clearance (the stability of MR-proADM is at least 75 days in the absence of clinical changes). Therefore, it can only be measured once at the time of patient hospitalization [68].

### 4.6. Statistical Analysis

To ensure the diagnostic, prognostic and prevalence accuracy of organ damage, we evaluated the sensitivity, specificity, predictive value and likelihood ratio of MR-proADM.

To this end, univariate and multivariate analyses were performed to evaluate the correlation of MR-proADM values at admission above the cut-off of 1 ng/mL in patients with localized infection and above the cut-off of 2 ng/mL and 3 ng/mL in septic patients, because these were the ones to which the best sensitivity, specificity and likelihood ratio values corresponded, chosen on the basis of recent meta-analyses and also our previous studies [7,16,46,47,48,49].

All continuous laboratory and clinical variables (of septic patients with or without septic shock, and of septic patients vs. control patients) were compared using the non-parametric Mann–Whitney test, and the results are represented as the median and interquartile range (i.e., 25–75th percentile, IQR). Categorical variables are reported as counts and percentages and were assessed by chi-square or Fisher’s exact tests. A *p* value < 0.05 were considered significant.

An AUROC of 0.5 was considered non-predictive and 1.0 was considered to indicate perfect predictive ability. An AUROC of 0.70 to 0.80 was considered acceptable.

Receiver operating characteristic (ROC) analysis was performed among independent variables associated with organ damage and mortality to define the optimal cut-off point for plasma MR-proADM.

Areas under the curve (AUCs) and their significance were calculated. The χ^2^ for proportions test was used to compare the relative percentages of prevalence in the patient groups comparison. A *p* value < 0.05 was considered significant. Stepwise multiple logistic regression was used for multivariate analysis, using, as dependent variables, MR-proADM above the cut-off points, 30-day and 90-day mortality and the following independent variables: AKI, ARDS, acute liver failure, AHF, RIFLE class, ARDS group, GCS < 15, the need for ICU transfer, the need for catecholamine administration, q-SOFA and SOFA scores ≥ 2, shock and multiple organ involvement.

Odd ratios were computed and their significance reported. All probabilities were two-tailed, and *p* values ≤ 0.05 were regarded as significant [7,17]. The MedCalc statistical package was used for statistical analysis (MedCalc^®^ Statistical Software version 22.006 (MedCalc Software Ltd., Ostend, Belgium; https://www.medcalc.org (Version 22.016); 2023), as well as IBM SPSS Statistics version 27.0.1.0.

## 5. Conclusions

MR-proADM is a marker of sepsis that reflects the level of oxidative stress and, thus, the severity of the disease proportionally to organ damage.

The value of MR-proADM is directly proportional to the severity of the organ damage and the prognosis.

MR-proADM identifies patients who have one or more organ failures, even if the correlation is stronger for some variables (AKI, GCS < 15, need for ICU transfer) that reflect more severe outcomes.

The clinical use of MR-proADM is to identify the phenotype of septic patients at the greatest risk of potentially lethal organ damage and death, by identifying patients who need early and intensive therapeutic treatment.

## Figures and Tables

**Figure 1 ijms-24-17429-f001:**
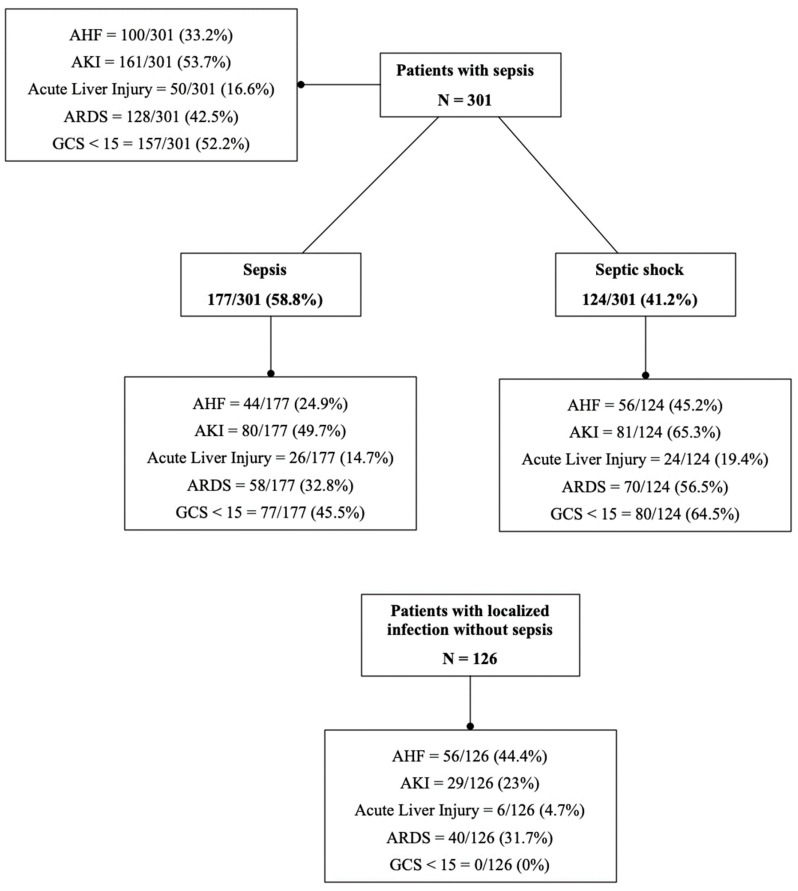
Stratification of organ damage in septic patients and those with localized infection.

**Figure 2 ijms-24-17429-f002:**
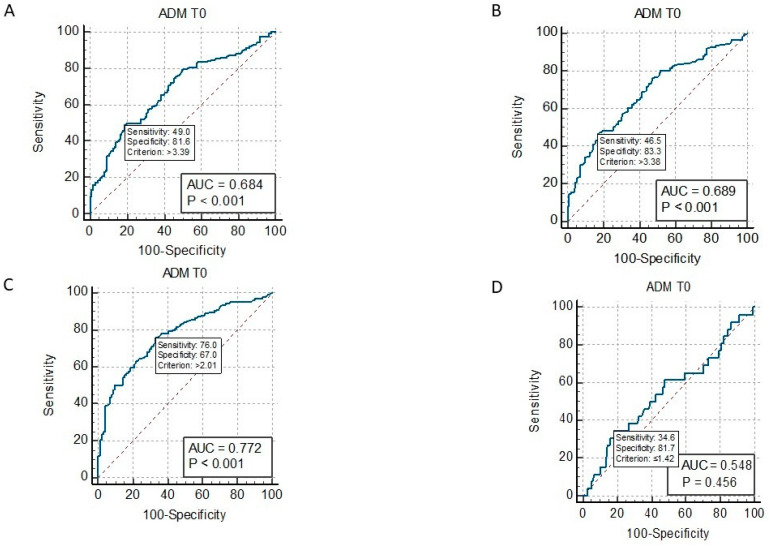
ROC curve of MR-proADM at admission (ADM T0) and AKI in septic patients (panel (**A**)); ROC curve of MR-proADM at admission (ADM T0) and multiorgan failure in septic patients (panel (**B**)); ROC curve of MR-proADM at admission (ADM T0) and SOFA score ≥ 2 (panel (**C**)); and ROC curve of MR-proADM at admission (ADM T0) and q-SOFA score ≥ 2 (panel (**D**)).

**Figure 3 ijms-24-17429-f003:**
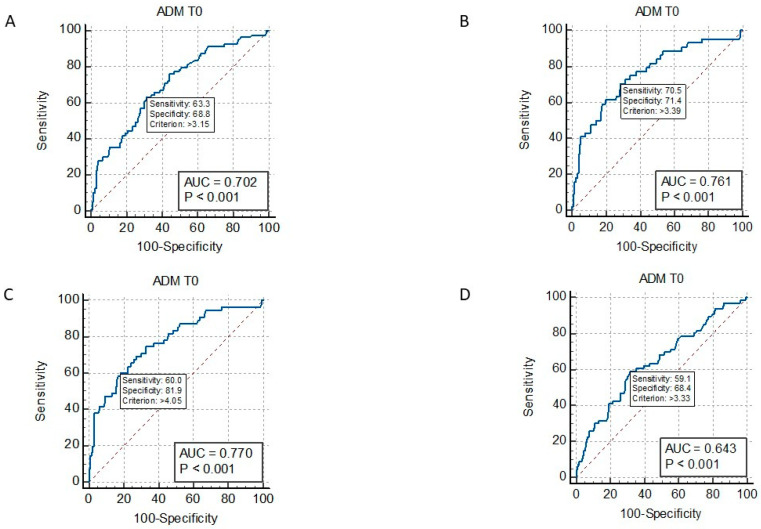
ROC curve of MR-proADM at admission (ADM T0) and mortality in septic patients (panel (**A**)); ROC curve of MR-proADM at admission (ADM T0) and 30-day mortality in septic patients (panel (**B**)); ROC curve of MR-proADM at admission (ADM T0) and 90-day mortality in septic patients (panel (**C**)); and ROC curve of MR-proADM at admission (ADM T0) and need for ICU transfer in septic patients (panel (**D**)).

**Table 1 ijms-24-17429-t001:** Demographic characteristics, clinical scores and inflammatory biomarkers of the study population classified as patients with sepsis and septic shock and patients with infection without sepsis. * Comparison between patients with sepsis and patients with septic shock. ** Comparison between all patients with sepsis and patients with infection without sepsis.

Variables	Patients with Sepsis ** N = 301				Patients with Infection without Sepsis ** N = 126	
	Total	Sepsis * N = 177	Septic shock * N = 124	*p* *		*p* **
Age in years, median [IQR]	74.5 [66.082.0]	73 [64.0–81.0]	77 [68.0–83.0]	0.019	79 [69.8–86.0]	0.0014
Sex category, male (%)	164 (54%)	97 (54%)	67 (54%)	1	63 (50%)	0.45
**Anamnestic variables, n (%)**						
Cancer	24 (8%)	17 (9.6%)	7 (5.6%)	0.20	27 (21.4%)	0.0001
COPD	51 (17%)	31 (17.5%)	20 (16%)	0.73	38 (30.2%)	0.002
Cardiovascular disease	113 (37.5%)	56 (31.6%)	57 (46%)	0.0017	68 (54%)	0.011
Liver disease	24 (8%)	10 (5.6%)	14 (11.3%)	0.072	7 (5.6%)	0.62
Chronic kidney disease	52 (17.3%)	25 (14.2%)	27 (21.8%)	0.087	33 (26.2%)	0.36
Diabetes mellitus	51 (17%)	36 (20.3%)	15 (12%)	0.0001	27 (21.4%)	0.28
Cerebrovascular disease	32 (10.6%)	20 (11.3%)	12 (9.7%)	0.65	68 (54%)	<0.0001
SOFA ≥ 2	252 (83.7%)	140 (79%)	112 (90.3%)	0.009	71 (56%)	<0.0001
q-SOFA ≥ 2	21 (7%)	7 (4%)	14 (11.3%)	0.015	1 (0.8%)	0.0084
PCT, median [IQR]	1.4 [0.37–6.7]	0.97 [0.3–4.7]	2.5 [0.7–12.8]	0.0001	0.10 [0.05–0.3]	<0.0001
MR-proADM, median [IQR]	2.5 [1.6–4.3]	2.12 [1.4–3.2]	3.65 [2.0–6.4]	<0.0001	1.19 [0.84–1.87]	<0.0001
Acute heart failure (%)	100 (33.2%)	44 (24.9%)	56 (45.2%)	0.0002	56 (44.4%)	0.028
AKI (%)	161 (53.7%)	80 (49.7%)	81 (65.3%)	0.0074	29 (23%)	<0.0001
Acute liver injury (%)	50 (16.6%)	26 (14.7%)	24 (19.4%)	0.28	6 (4.7%)	0.0009
ARDS (%)	128 (42.5%)	58 (32.8%)	70 (56.5%)	<0.0001	40 (31.7%)	0.037
GCS < 15 (%)	157 (52.2%)	77 (45.5%)	80 (64.5%)	0.0012	0 (0%)	<0.0001
**Organ damage**						
Score 0	38 (12.6%)	28 (15.8%)	10 (8%)	0.045	25 (19.8%)	0.056
Score 1	78 (26%)	63 (35.6%)	15 (12%)	<0.0001	41 (32.5%)	0.172
Score ≥ 2	185 (61.4%)	86 (48.6%)	99 (80%)	<0.0001	60 (47.6%)	0.0086
MR-proADM ≥ 2	168 (55.8%)	83(46.9%)	86 (69.3%)	0.0001	29 (23.0%)	<0.0001
MR-proADM ≥ 3	111(36.8%)	46 (25.9%)	66 (53.2%)	<0.0001	11 (8.73%)	<0.0001
Need of ICU transfer, n (%)	66 (21.9%)	28 (15.8%)	38 (30.6)	0.0023	0 (0%)	<0.0001
LOS median [IQR])	15 [10.5–26.5]	15 [8.2–36.2]	17 [11.0–26.0]	0.83	10 [8.0–13.0]	<0.0001
Mortality, n (%)	79 (26.2%)	27 (15.3%)	52 (42%)	<0.0001	0 (0%)	<0.0001

AST, aspartate transaminase; ALT, alanine transaminase; ARDS, acute respiratory distress syndrome; BMI, body mass index; COPD, chronic obstructive pulmonary disease; ICU, intensive care unit; INR, international normalized ratio; IQR, interquartile range; GCS, Glasgow Coma Scale; LOS, length of stay; MR-proADM mid-regional pro-adrenomedullin; PCT, procalcitonin; q-SOFA, quick Sequential Organ Failure Assessment; RIFLE, Risk, Injury, Failure, Loss of Kidney Function and End-Stage Kidney Disease; SOFA, Sequential Organ Failure Assessment.

**Table 2 ijms-24-17429-t002:** Univariate analysis: odds ratio, interval of confidence (IC) and statistical significance between MR-proADM value measured at admission ≥ 2 ng/mL or ≥3 ng/mL and clinical variables.

**MR-proADM ≥ 2 ng/mL**	**Odds Ratio**	**IC**	** *p* **
Multiple organ failure	3.2964	2.0094 to 5.4078	<0.0001
AKI	3.4857	2.1153 to 5.7439	<0.0001
Anemia	2.4559	1.1582 to 5.2073	0.019
GCS < 15	2.0878	1.2897 to 3.3798	0.0027
ARDS	1.8679	1.1407 to 3.0587	0.013
ICU transfer	1.9091	1.0384 to 3.5096	0.037
Need for catecholamine	2.5276	1.4197 to 4.5001	0.001
q-SOFA score ≥ 2	ns *	ns	ns
SOFA score ≥ 2	5.6863	3.3395 to 9.6821	<0.0001
Septic shock	2.4959	1.4965 to 4.1626	0.0005
AHF	1.8506	1.0947 to 3.1284	0.022
Acute liver failure	ns	ns	ns
**MR-proADM ≥ 3 ng/mL**	**Odds ratio**	**IC**	** *p* **
Multiple organ failure	2.2130	1.3772 to 3.5559	0.0010
AKI	2.0689	1.3052 to 3.2793	0.0020
Anemia	2.3646	1.0731 to 5.2104	0.0328
GCS < 15	2.2720	1.4316 to 3.6056	0.0005
ARDS	1.8222	1.1469 to 2.8950	0.0111
ICU transfer	ns	ns	ns
Need for catecholamine	2.7650	1.6373 to 4.6693	0.0001
q-SOFA score ≥ 2	ns	ns	ns
SOFA score ≥ 2	3.9315	2.3094 to 6.6930	<0.0001
Septic shock	2.4435	1.5235 to 3.9192	0.0002
AHF	ns	ns	ns
Acute liver failure	ns	ns	ns

* ns: not significant.

**Table 3 ijms-24-17429-t003:** Multivariate logistic regression analysis: MR-proADM value measured at admission ≥ 2 pg/mL or ≥3 ng/mL and clinical variables.

**MR-proADM ≥ 2 ng/mL**	**Odds Ratio**	**IC**	** *p* **
Multiple organ failure	ns *	ns	ns
AKI	3.3300	1.7301 to 6.4093	0.0003
Anemia	2.5228	1.0561 to 6.0265	0.0373
GCS	ns	ns	ns
ARDS	ns	ns	ns
ICU transfer	ns	ns	ns
Need for catecholamine	ns	ns	ns
q-SOFA score ≥ 2	ns	ns	ns
SOFA score ≥ 2	4.1487	2.2255 to 7.7341	<0.0001
Septic shock	ns	ns	ns
AHF	ns	ns	ns
Acute liver failure	ns	ns	ns
**MR-proADM ≥ 3 ng/mL**	**Odds ratio**	**IC**	** *p* **
Multiple organ failure	ns	ns	ns
AKI	2.1161	1.1605 to 3.8586	0.0145
Anemia	ns	ns	ns
GCS < 15	2.1025	0.8989 to 4.9174	0.034
ARDS	ns	ns	ns
ICU transfer	ns	ns	ns
Need for catecholamine	ns	ns	ns
q-SOFA score ≥ 2	ns	ns	ns
SOFA score ≥ 2	2.4078	1.3021 to 4.4526	0.0051
Septic shock	ns	ns	ns
AHF	ns	ns	ns
Acute liver failure	ns	ns	ns

* ns: not significant.

**Table 4 ijms-24-17429-t004:** Univariate analysis: 30-day mortality and 90-day mortality with clinical variables and laboratory parameters.

**30-Day Mortality**	**Odds Ratio**	**IC**	** *p* **
Multiple organ failure	3.2617	1.4581 to 7.2966	0.0040
AKI	2.3622	1.1786 to 4.7343	0.0154
Anemia	ns *	ns	ns
GCS < 15	5.0579	2.2629 to 11.3050	0.0001
ARDS	2.4531	1.2724 to 4.7294	0.0074
ICU transfer	4.9762	2.5408 to 9.7461	<0.0001
Need for cathecolamine	7.3522	3.6562 to 14.7843	<0.0001
q-SOFA score ≥ 2	ns	ns	ns
SOFA score ≥ 2	11.3174	2.6773 to 47.8394	0.0010
Septic shock	6.3089	2.9794 to 13.3589	<0.0001
AHF	ns	ns	ns
Liver failure	ns	ns	ns
MR-proADM ≥ 2 ng/mL	3.9317	1.7715 to 8.7264	0.0008
MR-proADM ≥ 3 ng/mL	3.9429	1.8690 to 8.3179	0.0003
**90-Day Mortality**	**Odds Ratio**	**IC**	** *p* **
Multiple organ failure	4.5778	2.1360 to 9.8108	0.0001
AKI	2.5213	1.3388 to 4.7481	0.0042
Anemia	ns	ns	ns
GCS <15	6.2588	3.0631 to 12.7888	<0.0001
ARDS	2.7929	1.5179 to 5.1387	0.0010
ICU transfer	5.8121	3.0097 to 11.2242	<0.0001
Need for catecholamine	4.7885	2.5684 to 8.9276	<0.0001
q-SOFA score ≥ 2	ns	ns	ns
SOFA score ≥ 2	15.9779	3.7925 to 67.3158	0.0002
Septic shock	3.8559	2.0438 to 7.2744	<0.0001
AHF	ns	ns	ns
Acute liver failure	ns	ns	ns
MR-proADM ≥ 2 ng/mL	3.9317	1.7715 to 8.7264	0.0008
MR-proADM ≥ 3 ng/mL	3.6742	1.8678 to 7.2276	0.0002

* ns: not significant.

**Table 5 ijms-24-17429-t005:** Multivariate logistic regression analysis: 30-day mortality and 90-day mortality with clinical variables and laboratory parameters.

**30-Day Mortality**	**Odds Ratio**	**IC**	** *p* **
Multiple organ failure	ns *	ns	ns
AKI	3.4656	1.2347 to 9.7274	0.0183
Anemia	ns	ns	ns
GCS < 15	5.1476	1.6979 to 15.6061	0.0038
ARDS	ns	ns	ns
ICU transfer	4.6601	1.9889 to 10.9190	0.0004
Need for catecholamine	4.3769	1.4054 to 13.6316	0.0109
q-SOFA score ≥ 2	ns	ns	ns
SOFA score ≥ 2	ns	ns	ns
Septic shock	ns	ns	ns
AHF	ns	ns	ns
Acute liver failure	ns	ns	ns
MR-proADM ≥ 2 ng/mL	ns	ns	ns
MR-proADM ≥ 3 ng/mL	ns	ns	ns
**90-Day Mortality**	**Odds Ratio**	**IC**	** *p* **
Multiple organ failure	ns	ns	ns
AKI	3.7243	1.4342 to 9.6714	0.0069
Anemia	ns	ns	ns
GCS < 15	4.2173	1.5692 to 11.3338	0.0043
ARDS	ns	ns	ns
ICU transfer	6.1198	2.5688 to 14.5795	<0.0001
Need for catecholamine	ns	ns	ns
q-SOFA score ≥ 2	ns	ns	ns
SOFA score ≥ 2	ns	ns	ns
Septic shock	ns	ns	ns
AHF	ns	ns	ns
Acute liver failure	ns	ns	ns
MR-proADM ≥ 2 ng/mL	ns	ns	ns
MR-proADM ≥ 3 ng/mL	ns	ns	ns

* ns: not significant.

**Table 6 ijms-24-17429-t006:** Multivariate logistic regression analysis: 30-day mortality and 90-day mortality with clinical variables and laboratory parameters, including the model stratification of patients with AKI by RIFLE criteria and ARDS severity.

**30-Day Mortality**	**Odds Ratio**	**IC**	** *p* **
Multiple organ failure	0.1366	0.0247 to 0.7550	0.0225
AKI	3.2293	1.0380 to 10.0464	0.0429
RIFLE (E and F)	3.3287	1.0041 to 11.0349	0.0492
Anemia	ns *	ns	ns
GCS < 15	4.9128	1.4870 to 16.2308	0.0090
ARDS	ns	ns	ns
ARDS grade 2 and 3	ns	ns	ns
ICU transfer	7.1586	2.6683 to 19.2054	0.0001
Need for catecholamine	3.9473	1.0845 to 14.3680	0.0373
q-SOFA score ≥ 2	ns	ns	ns
SOFA score ≥ 2	ns	ns	ns
Septic shock	ns	ns	ns
AHF	ns	ns	ns
Acute liver failure	ns	ns	ns
**90-Day Mortality**	**Odds Ratio**	**IC**	** *p* **
Multiple organ failure	ns	ns	ns
AKI	3.4156	1.2496 to 9.3360	0.0166
RIFLE (E and F)	ns	ns	ns
Anemia	ns	ns	ns
GCS < 15	4.3246	1.5959 to 11.7187	0.0040
ARDS	ns	ns	ns
ARDS grade 2 and 3	ns	ns	ns
ICU transfer	6.2453	2.5989 to 15.0078	<0.0001
Need for catecholamine	ns	ns	ns
q-SOFA score ≥ 2	ns	ns	ns
SOFA score ≥ 2	6.2167	1.2449 to 31.0450	0.0260
Septic shock	ns	ns	ns
AHF	ns	ns	ns
Acute liver failure	ns	ns	ns

* ns: not significant.

**Table 7 ijms-24-17429-t007:** Univariate analysis: correlation between MR-proADM ≥1.4 ng/mL in patients with localized infection and clinical parameters.

MR-proADM ≥ 1.4 ng/mL	Odds Ratio	IC	*p*
AKI	11.9111	4.1181 to 34.4513	<0.0001
Anemia	4.3343	1.8871 to 9.9551	0.0005
ARDS	ns *	ns	ns
q-SOFA score ≥ 2	ns	ns	ns
SOFA score ≥ 2	2.7206	1.2037 to 6.1492	0.0161
AHF	4.1355	1.9212 to 8.9023	0.0003
Acute liver failure	ns	ns	ns

* ns: not significant.

**Table 8 ijms-24-17429-t008:** Multivariate logistic regression analysis: correlation between MR-proADM ≥ 1.4 ng/mL in patients with localized infection and clinical parameters.

MR-proADM ≥ 1.4 ng/mL	Odds Ratio	IC	*p*
AKI	5.2434	1.6331 to 16.8349	0.0054
Anemia	2.8312	1.0660 to 7.5197	0.0368
ARDS	ns *	ns	ns
q-SOFA score ≥ 2	ns	ns	ns
SOFA score ≥ 2	1.9625	0.6955 to 5.5374	0.2027
AHF	3.2017	1.2562 to 8.1601	0.0148
Acute liver failure	ns	ns	ns

* ns: not significant.

## Data Availability

The data are contained within the article. The further research data presented in this study are available upon request from the corresponding author.

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
