# Peer review of "Mid-Regional Pro-Adrenomedullin Can Predict Organ Failure and Prognosis in Sepsis?"

_ijms, 2023, doi:10.3390/ijms242417429_

Round 1

Reviewer 1 Report

Comments and Suggestions for Authors

Spoto et al have assessed whether mid-regional pro-adrenomedullin (xx) can predicts organ failure and prognosis in sepsis in a retrospective analysis of 301 septic patients.

1)      Many studies on the predictive value of MR-pro-ADM have already been published. For instance, a recent meta-analysis: “Comparison of presepsin and Mid-regional pro-adrenomedullin in the diagnosis of sepsis or septic shock: a systematic review and meta-analysis. Liang J, Cai Y, Shao Y. BMC Infect Dis. 2023 May 5;23(1):288.” What is the added value of this study?

2)  In the introduction, the authors give a very causal role to the increase in ADM which is observed with sepsis. Causality cannot be derived from association data, please carefully rephrase this throughout the manuscript.

3)  There is no statistical plan on how to assess the predictive value of MR-pro-ADM.  The result section includes a very long and detailed description of the baseline characteristics,( information which is also presented in  Table 1). Where the authors describe differences between patients with septic shock, sepsis and isolated infection. What is the link with MR-pro-ADM?  How was the decision made to used cut-offs of 2 and 3 ng/ml?

4)    The discussion is very long and lacks focus

Comments on the Quality of English Language

can be improved

Author Response

Spoto et al have assessed whether mid-regional pro-adrenomedullin (xx) can predicts organ failure and prognosis in sepsis in a retrospective analysis of 301 septic patients.

  • Many studies on the predictive value of MR-pro-ADM have already been published. For instance, a recent meta-analysis: “Comparison of presepsin and Mid-regional pro-adrenomedullin in the diagnosis of sepsis or septic shock: a systematic review and meta-analysis. Liang J, Cai Y, Shao Y. BMC Infect Dis. 2023 May 5;23(1):288.” What is the added value of this study?

Reply: We thank the Reviewer for the manuscript revision we performed the correction requested as follows:

The added value of this study is stressed in the discussion.

  • In the introduction, the authors give a very causal role to the increase in ADM which is observed with sepsis. Causality cannot be derived from association data, please carefully rephrase this throughout the manuscript.

Reply: We modify the introduction, as requested, Lines 68-70; 106-108

  • There is no statistical plan on how to assess the predictive value of MR-pro-ADM.  The result section includes a very long and detailed description of the baseline characteristics,( information which is also presented in  Table 1). Where the authors describe differences between patients with septic shock, sepsis and isolated infection. What is the link with MR-pro-ADM?  How was the decision made to used cut-offs of 2 and 3 ng/ml?

Reply: Statistical analysis was performed as reported in material and methods section. For group comparison, Mann Whitney and chi-quadro for proportions were used. For decisional cut-off points ROC curve analysis was performed. We hope that this clarification can be the answer to Reviewer suggestion.

  • The discussion is very long and lacks focus

Reply: Discussion was improved and reduced.

Reviewer 2 Report

Comments and Suggestions for Authors

This study attempts to demonstrate the ability of Mid Regional Pro-Adrenomedulin  (PMRA) levels drawn on admission to predict the severity of inflammation, organ dysfunction and endothelitis in patients with sepsis and septic shock.  In addition the investigators demonstrate the predictive value of PRMA on clinical outcomes. The authors employ various statistical measures to demonstrate their case.  The paper suffers from some issues related to grammar and stylistic issues in the presentation that hampers their findings.  Despite this the paper is important and should be improved in order to publish.  If the tables show results, the findings can be shown there and only summarized in table. This will make manuscript less lengthy.  This a well done study however the grammar needs to be improved I will defer to the editors in terms of the extent of revision.

Minor changes

1.The p values of univariate analysis should be placed in the table (Table 1  

2. ROC curves should be generated for all variable Fig 2  and 3 only includes some of them the rest are just reported in text, this makes it somewhat difficult to follow.

3. What statistical program was used?.The authors should document this in the text.

Major

1.      Three groups were compared was Kruskal wallis used as the non parametric stat rather than Mann Whitney in my opinion Kruskal Wallis would be more appropriate.

2.      What multivariate analysis did the the investigators use.  This should be stated i.e forward stepwise regression or other type.

3.      3  I find it hard to comprehend how patients with septic shock are transferred to ICU the authors should explain this.  Dis they have shock?

4.      The proportion of patients with the cut-offs PMRA levels employed in the the multivariate analysis should be shown in the univariate table and analyzed with Chi squares i.e.

Proportion of PRMA >/= 3 , >/=2 in patients with Sepsis Septic shock and localized infection

Comments on the Quality of English Language

Grammar needs to be improved in order to improve the impact of paper I will defer to editors. The manuscript was not numbered It is difficult for me to specify issues.  I defer to the editors.

Author Response

This study attempts to demonstrate the ability of Mid Regional Pro-Adrenomedulin  (PMRA) levels drawn on admission to predict the severity of inflammation, organ dysfunction and endothelitis in patients with sepsis and septic shock.  In addition the investigators demonstrate the predictive value of PRMA on clinical outcomes. The authors employ various statistical measures to demonstrate their case.  The paper suffers from some issues related to grammar and stylistic issues in the presentation that hampers their findings.  Despite this the paper is important and should be improved in order to publish.  If the tables show results, the findings can be shown there and only summarized in table. This will make manuscript less lengthy.  This a well done study however the grammar needs to be improved I will defer to the editors in terms of the extent of revision.

Minor changes

We thank the Reviewer for the manuscript revision we performed the correction requested as follows:

1.The p values of univariate analysis should be placed in the table (Table 1 )

Reply: p values were included for each variable of Table 1.

  1. ROC curves should be generated for all variable Fig 2  and 3 only includes some of them the rest are just reported in text, this makes it somewhat difficult to follow.

Reply: figure 2 and 3 were updated with all ROC curves

  1. What statistical program was used?.The authors should document this in the text.

Reply: Information about statistical software used have been included in the manuscript (MedCalc software, IBM SPSS Statistics version 27

Major

  1. Three groups were compared was Kruskal wallis used as the non parametric stat rather than Mann Whitney in my opinion Kruskal Wallis would be more appropriate.

Reply: As we stated in the results, the study population was composed of 301 patients affected by septic syndrome, subclassified in 177 patients with sepsis and 124 patients with septic shock, and of 126 control patients with localized infection without sepsis. However, we did not want to compare all these groups together. On the contrary, we wanted to compare patients with sepsis vs. patients with septic shock, and all the septic patients (301) vs. the 126 control patients without sepsis. For that reason, we used the Mann Whitney test because in both cases we compared two groups. These analyses are now better specified in Table 1 and in the statistical methods section.

  1. What multivariate analysis did the the investigators use.  This should be stated i.e forward stepwise regression or other type.

Reply: Stepwise multiple logistic regression

  1. I find it hard to comprehend how patients with septic shock are transferred to ICU the authors should explain this.  Dis they have shock?

Reply: Patients defined as septic shock were in shock but they were transferred to ICU if they need mechanic ventilation and/or plasma ultrafiltration. The information was added to the manuscript.

  1. The proportion of patients with the cut-offs PMRA levels employed in the the multivariate analysis should be shown in the univariate table and analyzed with Chi squares i.e.

Proportion of MR-proADM >/= 3 , >/=2 in patients with Sepsis Septic shock and localized infection

Reply: the information was added within table 1

Comments on the Quality of English Language

Grammar needs to be improved in order to improve the impact of paper I will defer to editors. The manuscript was not numbered It is difficult for me to specify issues.  I defer to the editors.

Reply: English language was revised.

Reviewer 3 Report

Comments and Suggestions for Authors

In this retrospective single center study, Spoto and colleagues aimed to assess the potential of mid-regional pro-adrenomedullin (MR-proADM) in predicting organ failure and prognosis in sepsis.  Using correlation and multivariate analyses, they detected correlations between MR-proADM and multi-organ failure, SOFA scores, acute kidney injury, ICU transfer, and anemia using various MR-proADM cutoff values. I have the following comments for the authors` consideration:

(1) Unfortunately, the findings do not allow establishing clear guidelines for the use of MR-proADM, for various cut-off values were used for the analyses. It is unclear how these values have been chosen and validated.

(2) A major limitation of the study is its design. Biomarker research requires two cohorts, one to establish cut-off values (“discovery arm”) and the second larger cohort to validate the findings.

(3) The authors` main suggestion appears to change throughout the text. MR-proADM is described as a marker of infection in the Abstract, whereas as a marker of oxidative stress in Conclusions. Which one is correct? Were MR-proADM levels assessed at different time points (e.g. at admission and discharge from ICU)? Were these values stable?

(4) The control group does not seem to be appropriate, for it consisted of older patients who had higher frequency of cancer, chronic kidney disease, cerebrovascular disease and diabetes. How were these potentially confounding factors taken into consideration in establishing MD-proADM cutoff values?

(5) Some of the correlations are difficult to interpret. For instance, MR-proADM correlated with anemia at cut-off value >2 ng/ml, but not at >3 ng/ml, whereas correlation with GCS showed the opposite.

(6) What would be the outcome if the authors applied the cut-off value (>1.4 ng/ml) used for the non-septic group to the sepsis or septic shock groups?

(7)  The Discussion reads like a review on MR-proADM, not a critical comparison of the results obtained with those of other biomarkers. What is the relevance of Fig.4, “sepsis endothelitiis” to sepsis, which is currently defined as a dysregulated immune response to infection or injury?

(8) What is the relevance of supplementary figures? Suppl Fig 1 depicts viral (presumably SARS-CoV2) infection, Suppl Fig 2 shows mechanisms that might be linked to “endothelitiis”. It is unclear how this latter would lead to increased CRP levels or how the effects of Ang 1-9 (anti-inflammatory, antiproliferative, vasodilatation, etc.) could be linked to tissue damage.

Comments on the Quality of English Language

The mansucript will benefit from editing for English.

Author Response

In this retrospective single center study, Spoto and colleagues aimed to assess the potential of mid-regional pro-adrenomedullin (MR-proADM) in predicting organ failure and prognosis in sepsis.  Using correlation and multivariate analyses, they detected correlations between MR-proADM and multi-organ failure, SOFA scores, acute kidney injury, ICU transfer, and anemia using various MR-proADM cutoff values. I have the following comments for the authors` consideration:

We thank the Reviewer for the manuscript revision we performed the correction requested as follows:

  • Unfortunately, the findings do not allow establishing clear guidelines for the use of MR-proADM, for various cut-off values were used for the analyses. It is unclear how these values have been chosen and validated.

Reply: Cut- off values were chosen based on ROC curve analysis.

  • A major limitation of the study is its design. Biomarker research requires two cohorts, one to establish cut-off values (“discovery arm”) and the second larger cohort to validate the findings.

Reply: we added this important limitation in discussion section.

  • The authors` main suggestion appears to change throughout the text. MR-proADM is described as a marker of infection in the Abstract, whereas as a marker of oxidative stress in Conclusions. Which one is correct? Were MR-proADM levels assessed at different time points (e.g. at admission and discharge from ICU)? Were these values stable?

Reply: MR-proADM is a marker of oxidative stress that can be consequent also to an infection. Regarding MR-proADM measurement, this was performed only at admission (T=0), since the marker has a slow clearance (the stability of MR-proADM is at least 75 days in the absence of clinical changes. Therefore, it can only be measured once at the time of patient hospitalization [Morgenthaler, N.G.; Struck, J.; Alonso, C.; Bergmann, A. Measurement of Midregional Proadrenomedullin in Plasma with an Immunoluminometric Assay. Clin. Chem. 2005, 51, 1823–1829]

  • The control group does not seem to be appropriate, for it consisted of older patients who had higher frequency of cancer, chronic kidney disease, cerebrovascular disease and diabetes. How were these potentially confounding factors taken into consideration in establishing MD-proADM cutoff values?

Reply: Control group was chosen as real-life patients matched for age sex as much as possible. For example age comparison in patients with septic shock and control group is not significant (p=0.89).

  • Some of the correlations are difficult to interpret. For instance, MR-proADM correlated with anemia at cut-off value >2 ng/ml, but not at >3 ng/ml, whereas correlation with GCS showed the opposite.

Reply: Correlation is significant at univariate analysis (table ) but it is not mantained at multivariate analysis probably because in this case anemia or GCS are influenced by the inclusion (stepwise) of the others variables.

(6) What would be the outcome if the authors applied the cut-off value (>1.4 ng/ml) used for the non-septic group to the sepsis or septic shock groups?

Reply: The cut-off of MR-proADM >1.4 ng/ml is lower in non- septic group because these patients are less critics than septic ones. In fact, MR-proADM cut-off in septic and septic shock patients is significantly higher. This higher cut-off is confirmed also in previous studies. 

(7)  The Discussion reads like a review on MR-proADM, not a critical comparison of the results obtained with those of other biomarkers. What is the relevance of Fig.4, “sepsis endothelitiis” to sepsis, which is currently defined as a dysregulated immune response to infection or injury?

Reply:   We thank the reviewer. The discussion was correct.

(8) What is the relevance of supplementary figures? Suppl Fig 1 depicts viral (presumably SARS-CoV2) infection, Suppl Fig 2 shows mechanisms that might be linked to “endothelitiis”. It is unclear how this latter would lead to increased CRP levels or how the effects of Ang 1-9 (anti-inflammatory, antiproliferative, vasodilatation, etc.) could be linked to tissue damage.

Reply: Suppl Figures were removed.

Comments on the Quality of English Language

The mansucript will benefit from editing for English.

Reply: English language was revised.

Round 2

Reviewer 1 Report

Comments and Suggestions for Authors

My questions 1,3 and 4 were not properly answered. 

1)    The added value of this study should be clear in the introduction. Now the authors list a number of studies already describing a the correlation of MR-proAD values with organ failure and outcome, so for the reader it remains a question why this study was done. To confirm previous findings? To add something?

1)     There is no statistical plan on how to assess the predictive value of MR-pro-ADM. Simply stating the tests that were used is not enough. The authors still first describe differences between patients with septic shock, sepsis and isolated infection in a very lenghty way. What is the link with MR-pro-ADM?  How was the decision made to used cut-offs of 2 and 3 ng/ml? he authors now state “Receiver operating characteristic (ROC) analysis was performed among independent variables associated with organ damage and mortality to define the optimal cut-off point for plasma MR-proADM.” But this generates a number of cut-offs as different variables were assessed. It is still not clear how than the decision was made to use 2 and 3 ng/ml as cut-off for the regression analysis.

2)     The discussion is still long and lacks focus. Results should not be placed in the discussion, while now the first 8 paragraphs are again descriptions of septic characteristics.

Comments on the Quality of English Language

Author Response

Reviewer 1

My questions 1,3 and 4 were not properly answered. 

-The added value of this study should be clear in the introduction. Now the authors list a number of studies already describing a the correlation of MR-proAD values with organ failure and outcome, so for the reader it remains a question why this study was done. To confirm previous findings? To add something?

Reply: We thank the reviewer for the possibility of clarifying the added value of our study, also in light of the recent meta-analyses.

The study of Adrenomedullin remains a key point, fundamental in clinical research with still many obscure points to be explored and many therapeutic benefits to be obtained.

We have therefore improved the aim, indicated the added value of our study, expanded the introduction and discussion, as follows:

- “In the literature there is no consensus on a value of MR-proADM that is diagnostic, prognostic or expresses specific organ damage, its quantification or the need for ICU transfer”.

Liang J, Cai Y, Shao Y. Comparison of presepsin and Mid-regional pro-adrenomedullin in the diagnosis of sepsis or septic shock: a systematic review and meta-analysis. BMC Infect Dis. 2023 May 5;23(1):288. doi: 10.1186/s12879-023-08262-4. PMID: 37147598; PMCID: PMC10160726.

Li P, Wang C, Pang S. The diagnostic accuracy of mid-regional pro-adrenomedullin for sepsis: a system­atic review and meta-analysis. Minerva Anestesiol 2021;87:1117-27. DOI: 10.23736/S0375-9393.21.15585-3)

Baldirà, J.; Ruiz-Rodríguez, J.C.; Ruiz-Sanmartin, A.; Chiscano, L.;Cortes, A.; Sistac, D.Á.; Ferrer-Costa,R.; Comas, I.; Villena, Y.; Larrosa,M.N.; et al. Use of Biomarkers to Improve 28-Day Mortality Stratification in Patients with Sepsis and SOFA _ 6. Biomedicines 2023, 11,2149. https://doi.org/10.3390/ biomedicines11082149

Angeletti S, Dicuonzo G, Fioravanti M, De Cesaris M, Fogolari M, Lo Presti A, et al. Procalcitonin, MR-Proadre­nomedullin, and Cytokines Measurement in Sepsis Diag­nosis: Advantages from Test Combination. Dis Markers 2015;2015:951532. 

Sargentini V, Collepardo D, D Alessandro M, Petralito G, Ceccarelli G, Alessandri F, et al. Role of biomarkers in adult sepsis and their application for a good laboratory practice: a pilot study. J Biol Regul Homeost Agents 2017;31:1147–54.

Spoto S, Nobile E, Carnà EPR, Fogolari M, Caputo D, De Florio L, Valeriani E, Benvenuto D, Costantino S, Ciccozzi M, Angeletti S. Best diagnostic accuracy of sepsis combining SIRS criteria or qSOFA score with Procalcitonin and Mid-Regional pro-Adrenomedullin outside ICU. Sci Rep. 2020 Oct 6;10(1):16605. doi: 10.1038/s41598-020-73676-y. PMID: 33024218; PMCID: PMC7538435.

Valenzuela-Sánchez F, Valenzuela-Méndez B, Bohollo de Austria R, Rodríguez-Gutiérrez JF, Estella-García Á, Fernández-Ruiz L, González-García MÁ, Rello J. Plasma levels of mid-regional pro-adrenomedullin in sepsis are associated with risk of death. Minerva Anestesiol. 2019 Apr;85(4):366-375. doi: 10.23736/S0375-9393.18.12687-3. Epub 2018 Sep 10. PMID: 30207133.)

-“Aim of the study is to determine an MR-proADM value that, in addition to clinical diagnosis, can identify patients with localized infection or those with sepsis/septic shock, with specific organ damage or with the need for ICU transfer and prognosis.

Secondary outcomes is to correlate MR-proADM value with length of stay (LOS)”.

 -“The added value of the study is to provide the clinician with an MR-proADM value which, in addition to the clinical infectious diagnosis, can identify patients with: a) localized infection or sepsis/septic shock, b) with specific organ damage, c) with the need for ICU transfer and d) with the prognosis, to be able to treat them as appropriately, promptly and intensively as possible, saving lives”.

- “The use of biomarkers assists in the clinical diagnosis of infection. Clinical sepsis scores allow septic patients to be stratified based on the number and severity of organ damage, therefore they become positive later than the elevation of MR-proADM which, moreover, can be affected by oxidative stress due to other even non-infectious causes. In our study the established cut off value of MR-proADM (with corresponding the best sensitivity and specificity value) to identify patients with localized infection was ≥ 1.44 ng/ml, to identify sepsis with organ damage was ≥ 2 ng/ml. The use of these cut-offs could indicate timely and intensive treatment, avoiding the onset of sepsis and organ damage and/or death. These results are in line with a recent meta-analysis and systematic review evaluated the diagnostic value of MR-proADM in sepsis, finding that MR-proADM is an excellent biomarker for the diagnosis of sepsis.

Li P, Wang C, Pang S. The diagnostic accuracy of mid-regional pro-adrenomedullin for sepsis: a system­atic review and meta-analysis. Minerva Anestesiol 2021;87:1117-27. DOI: 10.23736/S0375-9393.21.15585-3)

Baldirà, J.; Ruiz-Rodríguez, J.C.; Ruiz-Sanmartin, A.; Chiscano, L.;Cortes, A.; Sistac, D.Á.; Ferrer-Costa,R.; Comas, I.; Villena, Y.; Larrosa,M.N.; et al. Use of Biomarkers to Improve 28-Day Mortality Stratification in Patients with Sepsis and SOFA _ 6. Biomedicines 2023, 11,2149. https://doi.org/10.3390/ biomedicines11082149

Angeletti S, Dicuonzo G, Fioravanti M, De Cesaris M, Fogolari M, Lo Presti A, et al. Procalcitonin, MR-Proadre­nomedullin, and Cytokines Measurement in Sepsis Diag­nosis: Advantages from Test Combination. Dis Markers 2015;2015:951532. 

Sargentini V, Collepardo D, D Alessandro M, Petralito G, Ceccarelli G, Alessandri F, et al. Role of biomarkers in adult sepsis and their application for a good laboratory practice: a pilot study. J Biol Regul Homeost Agents 2017;31:1147–54.

Spoto S, Nobile E, Carnà EPR, Fogolari M, Caputo D, De Florio L, Valeriani E, Benvenuto D, Costantino S, Ciccozzi M, Angeletti S. Best diagnostic accuracy of sepsis combining SIRS criteria or qSOFA score with Procalcitonin and Mid-Regional pro-Adrenomedullin outside ICU. Sci Rep. 2020 Oct 6;10(1):16605. doi: 10.1038/s41598-020-73676-y. PMID: 33024218; PMCID: PMC7538435.

Valenzuela-Sánchez F, Valenzuela-Méndez B, Bohollo de Austria R, Rodríguez-Gutiérrez JF, Estella-García Á, Fernández-Ruiz L, González-García MÁ, Rello J. Plasma levels of mid-regional pro-adrenomedullin in sepsis are associated with risk of death. Minerva Anestesiol. 2019 Apr;85(4):366-375. doi: 10.23736/S0375-9393.18.12687-3. Epub 2018 Sep 10. PMID: 30207133.

In the sepsis population, using 1-1.5 nmol/L as the cut-off value of MR-proADM had a higher combined sensitivity and specificity for the diagnosis of sepsis, which were 0.83 and 0.90, respectively. Li P, Wang C, Pang S. The diagnostic accuracy of mid-regional pro-adrenomedullin for sepsis: a system­atic review and meta-analysis. Minerva Anestesiol 2021;87:1117-27. DOI: 10.23736/S0375-9393.21.15585-3)

Worthy of note is the essential aid that can provide use of a cut off of MR-proADM ≥ 1.5 nmol/L for early sepsis diagnosis in those with a negative SOFA score. In this study, indeed, about 35% of patients were negative for SIRS criteria or q-SOFA, and SOFA score or for all, despite evidence of positive blood culture and documented microbiological isolate or clinical diagnosis of infection. In these patients, the use of MR-proADM was crucial to provide early diagnosis and confirm the suspicion of sepsis. Spoto S, Nobile E, Carnà EPR, Fogolari M, Caputo D, De Florio L, Valeriani E, Benvenuto D, Costantino S, Ciccozzi M, Angeletti S. Best diagnostic accuracy of sepsis combining SIRS criteria or qSOFA score with Procalcitonin and Mid-Regional pro-Adrenomedullin outside ICU. Sci Rep. 2020 Oct 6;10(1):16605. doi: 10.1038/s41598-020-73676-y. PMID: 33024218; PMCID: PMC7538435.

Spoto, S.; Fogolari, M.; De Florio, L.; Minieri, M.; Vicino, G.; Legramante, J.; Lia, M.S.; Terrinoni, A.; Caputo, D.; Costantino, S.; et al. Procalcitonin and MR-proAdrenomedullin Combination in the Etiological Diagnosis and Prognosis of Sepsis and Septic Shock. Microbial Pathogenesis 2019, 137, 103763, doi:10.1016/j.micpath.2019.103763.

MR-proADM had a high accuracy in identifying both 28-day and 90-day mortality, compared to all other biomarkers and clinical scores.

Baldirà, J.; Ruiz-Rodríguez,J.C.; Ruiz-Sanmartin, A.; Chiscano, L.;Cortes, A.; Sistac, D.Á.; Ferrer-Costa,R.; Comas, I.; Villena, Y.; Larrosa, M.N.; et al. Use of Biomarkers to Improve 28-Day Mortality Stratification in Patients with Sepsis and SOFA _ 6. Biomedicines 2023, 11,

  1. https://doi.org/10.3390/biomedicines11082149

Valenzuela-Sánchez F, Valenzuela-Méndez B, Bohollo de Austria R, Rodríguez-Gutiérrez JF, Estella-García Á, Fernández-Ruiz L, González-García MÁ, Rello J. Plasma levels of mid-regional pro-adrenomedullin in sepsis are associated with risk of death. Minerva Anestesiol. 2019 Apr;85(4):366-375. doi: 10.23736/S0375-9393.18.12687-3. Epub 2018 Sep 10. PMID: 30207133.

Outside ICU, a MR-proADM cut-off values > 3.39 nmol/L in sepsis and > 4.33 nmol/L in septic shock were associated with significant higher risk of 90-days mortality. Spoto, S.; Fogolari, M.; De Florio, L.; Minieri, M.; Vicino, G.; Legramante, J.; Lia, M.S.; Terrinoni, A.; Caputo, D.; Costantino, S.; et al. Procalcitonin and MR-proAdrenomedullin Combination in the Etiological Diagnosis and Prognosis of Sepsis and Septic Shock. Microbial Pathogenesis 2019, 137, 103763, doi:10.1016/j.micpath.2019.103763.

In ICU patients, admitted with SIRS and organ dysfunction, an MR-proADM cut-off point of 1.425 nmol/L helps to identify those with sepsis, while an MR-proADM value above 5.626 nmol/L, 48 hours after admission, was associated with a high risk of death. Valenzuela-Sánchez F, Valenzuela-Méndez B, Bohollo de Austria R, Rodríguez-Gutiérrez JF, Estella-García Á, Fernández-Ruiz L, González-García MÁ, Rello J. Plasma levels of mid-regional pro-adrenomedullin in sepsis are associated with risk of death. Minerva Anestesiol. 2019 Apr;85(4):366-375. doi: 10.23736/S0375-9393.18.12687-3. Epub 2018 Sep 10. PMID: 30207133.

The added value of our study is in the clinical use of MR-proADM to individuate septic patients at higher risk of death by identifying who may also benefit from more strictly medical treatment including hemodynamic management, infection source control and hard and timely antibiotic therapy, modulation of host response therapy also with adrecizumab, with administration as early as possible upon evidence of sepsis with AKI, impaired GCS or shock [55-58].

Our study therefore, identifying the phenotype of the septic and infected patients with the worst prognosis, also responds to the necessary adherence to compliance with the "antimicrobial stewardship" rules to identify patients who need targeted or early empirical antibiotic therapy tailoring on the severity characteristics of the patient “[37, 55, 57, 59-62 ].

-There is no statistical plan on how to assess the predictive value of MR-pro-ADM. Simply stating the tests that were used is not enough. The authors still first describe differences between patients with septic shock, sepsis and isolated infection in a very lenghty way. Reply: We thank the reviewer for the possibility of clarifying the statistical plan used, included in the materials and methods section.

To express the best diagnostic, prognostic and prevalence accuracy of organ damage we evaluated the sensitivity, specificity, predictive value and like-hood ratio of MR-pro-ADM.

“To this end, univariate and multivariate analysis were performed to evaluate the correlation of MR-proADM values ​​at admission above the cut-off of 1 ng/mL in patients with localized infection and above the cut-off of 2 ng/mL and 3 ng/mL in septic patients, because they are those to which the best sensitivity, specificity and like-hood ratio values ​​correspond, chosen on the basis of recent meta-analyses and also our previous studies”.

Liang J, Cai Y, Shao Y. Comparison of presepsin and Mid-regional pro-adrenomedullin in the diagnosis of sepsis or septic shock: a systematic review and meta-analysis. BMC Infect Dis. 2023 May 5;23(1):288. doi: 10.1186/s12879-023-08262-4. PMID: 37147598; PMCID: PMC10160726.

Li P, Wang C, Pang S. The diagnostic accuracy of mid-regional pro-adrenomedullin for sepsis: a system­atic review and meta-analysis. Minerva Anestesiol 2021;87:1117-27. DOI: 10.23736/S0375-9393.21.15585-3)

Baldirà, J.; Ruiz-Rodríguez, J.C.; Ruiz-Sanmartin, A.; Chiscano, L.;Cortes, A.; Sistac, D.Á.; Ferrer-Costa,R.; Comas, I.; Villena, Y.; Larrosa,M.N.; et al. Use of Biomarkers to Improve 28-Day Mortality Stratification in Patients with Sepsis and SOFA _ 6. Biomedicines 2023, 11,2149. https://doi.org/10.3390/ biomedicines11082149

Angeletti S, Dicuonzo G, Fioravanti M, De Cesaris M, Fogolari M, Lo Presti A, et al. Procalcitonin, MR-Proadre­nomedullin, and Cytokines Measurement in Sepsis Diag­nosis: Advantages from Test Combination. Dis Markers 2015;2015:951532. 

Sargentini V, Collepardo D, D Alessandro M, Petralito G, Ceccarelli G, Alessandri F, et al. Role of biomarkers in adult sepsis and their application for a good laboratory practice: a pilot study. J Biol Regul Homeost Agents 2017;31:1147–54.

Spoto S, Nobile E, Carnà EPR, Fogolari M, Caputo D, De Florio L, Valeriani E, Benvenuto D, Costantino S, Ciccozzi M, Angeletti S. Best diagnostic accuracy of sepsis combining SIRS criteria or qSOFA score with Procalcitonin and Mid-Regional pro-Adrenomedullin outside ICU. Sci Rep. 2020 Oct 6;10(1):16605. doi: 10.1038/s41598-020-73676-y. PMID: 33024218; PMCID: PMC7538435.

Valenzuela-Sánchez F, Valenzuela-Méndez B, Bohollo de Austria R, Rodríguez-Gutiérrez JF, Estella-García Á, Fernández-Ruiz L, González-García MÁ, Rello J. Plasma levels of mid-regional pro-adrenomedullin in sepsis are associated with risk of death. Minerva Anestesiol. 2019 Apr;85(4):366-375. doi: 10.23736/S0375-9393.18.12687-3. Epub 2018 Sep 10. PMID: 30207133.)

  -What is the link with MR-pro-ADM?Reply: The biohumoral elevation of adrenomedullin dosed in the blood in response to a serious infection is a consequence of the teleological need of the organism to compensate for the damage caused by the infection. This has been demonstrated in vitro and in vivo with recent studies.

We explained as follows:

- “Thus, endotheliitis is the true cornerstone of the onset of organ damage and prognosis and not myocardial damage. Sepsis determines inflammation, oxidative stress, and endotheliitis which is expressed in damage to the endothelial barrier, reduction of antimicrobial properties, reduction of the vasodilating effect leading to edema, hemodynamic overload up to organ’s dysfunction. Recently, it has been demonstrated in vivo and in vitro how the endogenous peptide adrenomedullin serum levels increases in production and expression during severe infections because it is an efficient counter-regulatory molecule with the purpose of regulation and stabilization of the endothelial barrier, protection of the microcirculation and then of the hemodynamic balance”.

Temmesfeld-Wollbrück B, Hocke AC, Suttorp N, Hippenstiel S. Adrenomedullin and endothelial barrier function. Thromb Haemost. 2007 Nov;98(5):944-51. doi: 10.1160/th07-02-0128. PMID: 18000597.

Pittard AJ, Hawkins WJ, Webster NR. The role of the microcirculation in the multi-organ dysfunction syndrome. Clin Intensive Care. 1994;5(4):186–90.

Xie Z, Chen WS, Yin Y, Chan EC, Terai K, Long LM, et al. Adrenomedullin surges are linked to acute episodes of the systemic capillary leak syndrome (Clarkson disease). J Leukoc Biol. 2018;103(4):749–59.

Vigue B, Leblanc PE, Moati F, Pussard E, Foufa H, Rodrigues A, et al. Mid-regional pro-adrenomedullin (MR-proADM), a marker of positive

fluid balance in critically ill patients: results of the ENVOL study. Crit Care. 2016;20(1):363.

“MR-proADM has been proposed as a useful early diagnostic biomarker of serious infection, in critically ill patients, even in post-surgical states, and its concentrations correspond to microcirculatory and endothelial damage in the early stages of organ dysfunction before the development of organ failure and therefore also in patients with a low SOFA score”.

“Thus, endotheliitis is the true cornerstone of the onset of organ damage and prognosis and not myocardial damage. Sepsis determines inflammation, oxidative stress, and endotheliitis which is expressed in damage to the endothelial barrier, reduction of antimicrobial properties, reduction of the vasodilating effect leading to edema, hemodynamic overload up to organ’s dysfunction. Recently, it has been demonstrated in vivo and in vitro how the endogenous peptide adrenomedullin serum levels increases in production and expression during severe infections because it is an efficient counter-regulatory molecule with the purpose of regulation and stabilization of the endothelial barrier, protection of the microcirculation and then of the hemodynamic balance”.

Temmesfeld-Wollbrück B, Hocke AC, Suttorp N, Hippenstiel S. Adrenomedullin and endothelial barrier function. Thromb Haemost. 2007 Nov;98(5):944-51. doi: 10.1160/th07-02-0128. PMID: 18000597.

Pittard AJ, Hawkins WJ, Webster NR. The role of the microcirculation in the multi-organ dysfunction syndrome. Clin Intensive Care. 1994;5(4):186–90.

Xie Z, Chen WS, Yin Y, Chan EC, Terai K, Long LM, et al. Adrenomedullin surges are linked to acute episodes of the systemic capillary leak syndrome (Clarkson disease). J Leukoc Biol. 2018;103(4):749–59.

Vigue B, Leblanc PE, Moati F, Pussard E, Foufa H, Rodrigues A, et al. Mid-regional pro-adrenomedullin (MR-proADM), a marker of positive

fluid balance in critically ill patients: results of the ENVOL study. Crit Care. 2016;20(1):363.

 -How was the decision made to used cut-offs of 2 and 3 ng/ml? the authors now state “Receiver operating characteristic (ROC) analysis was performed among independent variables associated with organ damage and mortality to define the optimal cut-off point for plasma MR-proADM.” But this generates a number of cut-offs as different variables were assessed. It is still not clear how than the decision was made to use 2 and 3 ng/ml as cut-off for the regression analysis.  Reply: We thank the reviewer for the possibility to better define and clearly identify the MR-proADM cut-offs.We have better defined the aim, the added value of the study, and the sessions of the materials and methods, results, discussion and limits in this sense, as reported in the text as follows.

- “In the literature there is no consensus on a value of MR-proADM that is diagnostic, prognostic or expresses specific organ damage, its quantification or the need for ICU transfer”.

Liang J, Cai Y, Shao Y. Comparison of presepsin and Mid-regional pro-adrenomedullin in the diagnosis of sepsis or septic shock: a systematic review and meta-analysis. BMC Infect Dis. 2023 May 5;23(1):288. doi: 10.1186/s12879-023-08262-4. PMID: 37147598; PMCID: PMC10160726.

Li P, Wang C, Pang S. The diagnostic accuracy of mid-regional pro-adrenomedullin for sepsis: a system­atic review and meta-analysis. Minerva Anestesiol 2021;87:1117-27. DOI: 10.23736/S0375-9393.21.15585-3)

Baldirà, J.; Ruiz-Rodríguez, J.C.; Ruiz-Sanmartin, A.; Chiscano, L.;Cortes, A.; Sistac, D.Á.; Ferrer-Costa,R.; Comas, I.; Villena, Y.; Larrosa,M.N.; et al. Use of Biomarkers to Improve 28-Day Mortality Stratification in Patients with Sepsis and SOFA _ 6. Biomedicines 2023, 11,2149. https://doi.org/10.3390/ biomedicines11082149

Angeletti S, Dicuonzo G, Fioravanti M, De Cesaris M, Fogolari M, Lo Presti A, et al. Procalcitonin, MR-Proadre­nomedullin, and Cytokines Measurement in Sepsis Diag­nosis: Advantages from Test Combination. Dis Markers 2015;2015:951532. 

Sargentini V, Collepardo D, D Alessandro M, Petralito G, Ceccarelli G, Alessandri F, et al. Role of biomarkers in adult sepsis and their application for a good laboratory practice: a pilot study. J Biol Regul Homeost Agents 2017;31:1147–54.

Spoto S, Nobile E, Carnà EPR, Fogolari M, Caputo D, De Florio L, Valeriani E, Benvenuto D, Costantino S, Ciccozzi M, Angeletti S. Best diagnostic accuracy of sepsis combining SIRS criteria or qSOFA score with Procalcitonin and Mid-Regional pro-Adrenomedullin outside ICU. Sci Rep. 2020 Oct 6;10(1):16605. doi: 10.1038/s41598-020-73676-y. PMID: 33024218; PMCID: PMC7538435.

Valenzuela-Sánchez F, Valenzuela-Méndez B, Bohollo de Austria R, Rodríguez-Gutiérrez JF, Estella-García Á, Fernández-Ruiz L, González-García MÁ, Rello J. Plasma levels of mid-regional pro-adrenomedullin in sepsis are associated with risk of death. Minerva Anestesiol. 2019 Apr;85(4):366-375. doi: 10.23736/S0375-9393.18.12687-3. Epub 2018 Sep 10. PMID: 30207133.)

“Aim of the study is to determine an MR-proADM value that, in addition to clinical diagnosis, can identify patients with localized infection or those with sepsis/septic shock, with specific organ damage or with the need for ICU transfer and prognosis.

Secondary outcomes is to correlate MR-proADM value with length of stay (LOS)”.

2.4 Statistical analysis

To express the best diagnostic, prognostic and prevalence accuracy of organ damage we evaluated the sensitivity, specificity, predictive value and like-hood ratio of MR-pro-ADM.

To this end, univariate and multivariate analysis were performed to evaluate the correlation of MR-proADM values ​​at admission above the cut-off of 1 ng/mL in patients with localized infection and above the cut-off of 2 ng/mL and 3 ng/mL in septic patients, because they are those to which the best sensitivity, specificity and like-hood ratio values ​​correspond, chosen on the basis of recent meta-analyses and also our previous studies”.

Liang J, Cai Y, Shao Y. Comparison of presepsin and Mid-regional pro-adrenomedullin in the diagnosis of sepsis or septic shock: a systematic review and meta-analysis. BMC Infect Dis. 2023 May 5;23(1):288. doi: 10.1186/s12879-023-08262-4. PMID: 37147598; PMCID: PMC10160726.

Li P, Wang C, Pang S. The diagnostic accuracy of mid-regional pro-adrenomedullin for sepsis: a system­atic review and meta-analysis. Minerva Anestesiol 2021;87:1117-27. DOI: 10.23736/S0375-9393.21.15585-3)

Baldirà, J.; Ruiz-Rodríguez, J.C.; Ruiz-Sanmartin, A.; Chiscano, L.;Cortes, A.; Sistac, D.Á.; Ferrer-Costa,R.; Comas, I.; Villena, Y.; Larrosa,M.N.; et al. Use of Biomarkers to Improve 28-Day Mortality Stratification in Patients with Sepsis and SOFA _ 6. Biomedicines 2023, 11,2149. https://doi.org/10.3390/ biomedicines11082149

Angeletti S, Dicuonzo G, Fioravanti M, De Cesaris M, Fogolari M, Lo Presti A, et al. Procalcitonin, MR-Proadre­nomedullin, and Cytokines Measurement in Sepsis Diag­nosis: Advantages from Test Combination. Dis Markers 2015;2015:951532.  Controllare che non ci sia già

Sargentini V, Collepardo D, D Alessandro M, Petralito G, Ceccarelli G, Alessandri F, et al. Role of biomarkers in adult sepsis and their application for a good laboratory practice: a pilot study. J Biol Regul Homeost Agents 2017;31:1147–54.

Spoto S, Nobile E, Carnà EPR, Fogolari M, Caputo D, De Florio L, Valeriani E, Benvenuto D, Costantino S, Ciccozzi M, Angeletti S. Best diagnostic accuracy of sepsis combining SIRS criteria or qSOFA score with Procalcitonin and Mid-Regional pro-Adrenomedullin outside ICU. Sci Rep. 2020 Oct 6;10(1):16605. doi: 10.1038/s41598-020-73676-y. PMID: 33024218; PMCID: PMC7538435. Questo già c’è!

Valenzuela-Sánchez F, Valenzuela-Méndez B, Bohollo de Austria R, Rodríguez-Gutiérrez JF, Estella-García Á, Fernández-Ruiz L, González-García MÁ, Rello J. Plasma levels of mid-regional pro-adrenomedullin in sepsis are associated with risk of death. Minerva Anestesiol. 2019 Apr;85(4):366-375. doi: 10.23736/S0375-9393.18.12687-3. Epub 2018 Sep 10. PMID: 30207133.)

All continuous laboratory and clinical variables (of septic patients with or without septic shock, and of septic patients vs control patients) were compared using the non-parametric Mann-Whitney test  and the results were represented as median and interquartile range (i.e. 25th-75th percentile, IQR). Categorical variables are reported as counts and percentages and assessed by chi-square or Fisher exact tests. P value <0.05 were considered as significant.

Area under the receiver operating characteristic (AUROC) curves were used to identify the biomarker or clinical score with the greatest predictive value for each endpoint, with 95% confidence intervals (95% CI) compared to determine significance. Youden’s criterion established optimal cut-of values with corresponding sensitivity and specificity values.

An AUROC of 0.5 was considered non-predictive, and 1.0 was considered a perfect predictive ability. An AUROC of 0.70 to 0.80 was considered acceptable.

Receiver operating characteristic (ROC) analysis was performed among independent variables associated with organ damage and mortality to define the optimal cut-off point for plasma MR-proADM.

Areas under the curve (AUCs) and their significance were calculated. χ2 for proportions test was used to compare the relative percentage of prevalence in the patients’ groups comparison. p value < 0.05 were considered as significant. Stepwise multiple logistic regression was used for multivariate analysis using as dependent variables MR-proADM above cut-off points, 30-day and 90-day mortality and the following independent variables: AKI, ARDS, acute Liver failure, AHF, RIFLE class, ARDS Group, GCS < 15, need for ICU transfer, need for catecholamine administration, q-SOFA and SOFA score ≥ 2, shock, multiple organ involvement.

Odd ratio were computed and their significance reported. All probabilities were two-tailed, and p values ≤ 0.05 were regarded as significant [7,17]. MedCalc statistical package was used for statistical analysis (MedCalc® Statistical Software version 22.006 (MedCalc Software Ltd, Ostend, Belgium; https://www.medcalc.org ; 2023); IBM SPSS Statistics version 27.0.1.0

-The discussion is still long and lacks focus. Results should not be placed in the discussion, while now the first 8 paragraphs are again descriptions of septic characteristics.

Reply: We thank the Reviewer for the possibility of improving and expanding the discussion section.

We moved the relevant results to the appropriate section and added these parts, as follows:

“The use of biomarkers assists in the clinical diagnosis of infection. Clinical sepsis scores allow septic patients to be stratified based on the number and severity of organ damage, therefore they become positive later than the elevation of MR-proADM which, moreover, can be affected by oxidative stress due to other even non-infectious causes. In our study the established cut off value of MR-proADM (with corresponding the best sensitivity and specificity value) to identify patients with localized infection was ≥ 1.44 ng/ml, to identify sepsis with organ damage was ≥ 2 ng/ml. The use of these cut-offs could indicate timely and intensive treatment, avoiding the onset of sepsis and organ damage and/or death. These results are in line with a recent meta-analysis and systematic review evaluated the diagnostic value of MR-proADM in sepsis, finding that MR-proADM is an excellent biomarker for the diagnosis of sepsis”.

Li P, Wang C, Pang S. The diagnostic accuracy of mid-regional pro-adrenomedullin for sepsis: a system­atic review and meta-analysis. Minerva Anestesiol 2021;87:1117-27. DOI: 10.23736/S0375-9393.21.15585-3)

Baldirà, J.; Ruiz-Rodríguez, J.C.; Ruiz-Sanmartin, A.; Chiscano, L.;Cortes, A.; Sistac, D.Á.; Ferrer-Costa,R.; Comas, I.; Villena, Y.; Larrosa,M.N.; et al. Use of Biomarkers to Improve 28-Day Mortality Stratification in Patients with Sepsis and SOFA _ 6. Biomedicines 2023, 11,2149. https://doi.org/10.3390/ biomedicines11082149

Angeletti S, Dicuonzo G, Fioravanti M, De Cesaris M, Fogolari M, Lo Presti A, et al. Procalcitonin, MR-Proadre­nomedullin, and Cytokines Measurement in Sepsis Diag­nosis: Advantages from Test Combination. Dis Markers 2015;2015:951532. 

Sargentini V, Collepardo D, D Alessandro M, Petralito G, Ceccarelli G, Alessandri F, et al. Role of biomarkers in adult sepsis and their application for a good laboratory practice: a pilot study. J Biol Regul Homeost Agents 2017;31:1147–54.

Spoto S, Nobile E, Carnà EPR, Fogolari M, Caputo D, De Florio L, Valeriani E, Benvenuto D, Costantino S, Ciccozzi M, Angeletti S. Best diagnostic accuracy of sepsis combining SIRS criteria or qSOFA score with Procalcitonin and Mid-Regional pro-Adrenomedullin outside ICU. Sci Rep. 2020 Oct 6;10(1):16605. doi: 10.1038/s41598-020-73676-y. PMID: 33024218; PMCID: PMC7538435.

Valenzuela-Sánchez F, Valenzuela-Méndez B, Bohollo de Austria R, Rodríguez-Gutiérrez JF, Estella-García Á, Fernández-Ruiz L, González-García MÁ, Rello J. Plasma levels of mid-regional pro-adrenomedullin in sepsis are associated with risk of death. Minerva Anestesiol. 2019 Apr;85(4):366-375. doi: 10.23736/S0375-9393.18.12687-3. Epub 2018 Sep 10. PMID: 30207133.

In the sepsis population, using 1-1.5 nmol/L as the cut-off value of MR-proADM had a higher combined sensitivity and specificity for the diagnosis of sepsis, which were 0.83 and 0.90, respectively. Li P, Wang C, Pang S. The diagnostic accuracy of mid-regional pro-adrenomedullin for sepsis: a system­atic review and meta-analysis. Minerva Anestesiol 2021;87:1117-27. DOI: 10.23736/S0375-9393.21.15585-3)

Worthy of note is the essential aid that can provide use of a cut off of MR-proADM ≥ 1.5 nmol/L for early sepsis diagnosis in those with a negative SOFA score. In this study, indeed, about 35% of patients were negative for SIRS criteria or q-SOFA, and SOFA score or for all, despite evidence of positive blood culture and documented microbiological isolate or clinical diagnosis of infection. In these patients, the use of MR-proADM was crucial to provide early diagnosis and confirm the suspicion of sepsis. Spoto S, Nobile E, Carnà EPR, Fogolari M, Caputo D, De Florio L, Valeriani E, Benvenuto D, Costantino S, Ciccozzi M, Angeletti S. Best diagnostic accuracy of sepsis combining SIRS criteria or qSOFA score with Procalcitonin and Mid-Regional pro-Adrenomedullin outside ICU. Sci Rep. 2020 Oct 6;10(1):16605. doi: 10.1038/s41598-020-73676-y. PMID: 33024218; PMCID: PMC7538435.

Spoto, S.; Fogolari, M.; De Florio, L.; Minieri, M.; Vicino, G.; Legramante, J.; Lia, M.S.; Terrinoni, A.; Caputo, D.; Costantino, S.; et al. Procalcitonin and MR-proAdrenomedullin Combination in the Etiological Diagnosis and Prognosis of Sepsis and Septic Shock. Microbial Pathogenesis 2019, 137, 103763, doi:10.1016/j.micpath.2019.103763.

MR-proADM had a high accuracy in identifying both 28-day and 90-day mortality, compared to all other biomarkers and clinical scores.

Baldirà, J.; Ruiz-Rodríguez,J.C.; Ruiz-Sanmartin, A.; Chiscano, L.;Cortes, A.; Sistac, D.Á.; Ferrer-Costa,R.; Comas, I.; Villena, Y.; Larrosa, M.N.; et al. Use of Biomarkers to Improve 28-Day Mortality Stratification in Patients with Sepsis and SOFA _ 6. Biomedicines 2023, 11,

  1. https://doi.org/10.3390/biomedicines11082149

Valenzuela-Sánchez F, Valenzuela-Méndez B, Bohollo de Austria R, Rodríguez-Gutiérrez JF, Estella-García Á, Fernández-Ruiz L, González-García MÁ, Rello J. Plasma levels of mid-regional pro-adrenomedullin in sepsis are associated with risk of death. Minerva Anestesiol. 2019 Apr;85(4):366-375. doi: 10.23736/S0375-9393.18.12687-3. Epub 2018 Sep 10. PMID: 30207133.

Outside ICU, a MR-proADM cut-off values > 3.39 nmol/L in sepsis and > 4.33 nmol/L in septic shock were associated with significant higher risk of 90-days mortality. Spoto, S.; Fogolari, M.; De Florio, L.; Minieri, M.; Vicino, G.; Legramante, J.; Lia, M.S.; Terrinoni, A.; Caputo, D.; Costantino, S.; et al. Procalcitonin and MR-proAdrenomedullin Combination in the Etiological Diagnosis and Prognosis of Sepsis and Septic Shock. Microbial Pathogenesis 2019, 137, 103763, doi:10.1016/j.micpath.2019.103763.

In ICU patients, admitted with SIRS and organ dysfunction, an MR-proADM cut-off point of 1.425 nmol/L helps to identify those with sepsis, while an MR-proADM value above 5.626 nmol/L, 48 hours after admission, was associated with a high risk of death.

Valenzuela-Sánchez F, Valenzuela-Méndez B, Bohollo de Austria R, Rodríguez-Gutiérrez JF, Estella-García Á, Fernández-Ruiz L, González-García MÁ, Rello J. Plasma levels of mid-regional pro-adrenomedullin in sepsis are associated with risk of death. Minerva Anestesiol. 2019 Apr;85(4):366-375. doi: 10.23736/S0375-9393.18.12687-3. Epub 2018 Sep 10. PMID: 30207133.

Reviewer 2 Report

Comments and Suggestions for Authors

The revised manuscript has been reviewed. and is acceptable 

Author Response

Thank You for your revision.

Reviewer 3 Report

Comments and Suggestions for Authors

The authors have partially responded to my previous concerns. This would lead the reviewer no choice other than reiterate previous comments.

While cut off values were obtained by ROC curve analyses, it is still unclear how these were validated. The clinical utility of multiple cut-off values is questionable. Which one should be included in predicting organ failure?

The authors acknowledged the lack of validation of their findings using an independent cohort.  This represents a major limitation of the study and raises concerns about the clinical usefulness of the data obtained.

It is unclear why repeated measurements of MR-proADM cannot be performed. Would MR-proADM levels reflects changes in clinical symptoms?

What would be the outcome if the authors applied a cut-off value (e.g. <1.4 mg/ml) across the board to compare non-septic group and septic shock groups? In other words, would this cut-off distinguish these two groups?

In spite of the authors` statement that “the discussion was correct”, the Discussion lacks critical comparison of the findings with those in the published literature. For instance, how do the cut-off values compare and which one could then be used to guide diagnosis/prognosis?

 Limitations of control group (beyond age and sex) should also be discussed.

Comments on the Quality of English Language

The text will benefit from editing for English.

Author Response

Reviewer 3

-The authors have partially responded to my previous concerns. This would lead the reviewer no choice other than reiterate previous comments.

 Reply: We thank the Reviewer for the attention he is dedicating to the manuscript, we apologize for the superficiality of the initial response and we hope that he can appreciate the new version of the manuscript with the substantial and specific changes made.

-While cut off values were obtained by ROC curve analyses, it is still unclear how these were validated. The clinical utility of multiple cut-off values is questionable. Which one should be included in predicting organ failure?  Reply: We thank the reviewer for the possibility to better define and clearly identify the MR-proADM cut-offs. We have better defined the aim, the added value of the study, and the sessions of the materials and methods, results, discussion and limits in this sense, as reported in the text as follows. In particular, in septic patients we can use a cut off value of MR-proADM ≥ 2 ng/mL which identifies patients with ≥ 2 organ damages and which therefore corresponds to the SOFA score.

“In our study the established cut off value of MR-proADM (with corresponding the best sensitivity and specificity value) to identify patients with localized infection was ≥ 1.44 ng/ml, to identify patients with sepsis with SOFA score ≥ 2, with need for ICU transfer or not survived at 90-day was ≥ 2 ng/ml”.

-“ Aim of the study is to determine an MR-proADM value that, in addition to clinical diagnosis, can identify patients with localized infection or those with sepsis/septic shock, with specific organ damage or with the need for ICU transfer and prognosis.

Secondary outcomes is to correlate MR-proADM value with length of stay (LOS).

 “The added value of the study is to provide the clinician with an MR-proADM value which, in addition to the clinical diagnosis, can identify patients with: a) localized infection or sepsis/septic shock, b) with specific organ damage, c) with the need for ICU transfer and d) with the prognosis, to be able to treat them as appropriately, promptly and intensively as possible, saving lives”.

  • Statistical analysis.

To express the best diagnostic, prognostic and prevalence accuracy of organ damage we evaluated the sensitivity, specificity, predictive value and like-hood ratio of MR-pro-ADM.

To this end, univariate and multivariate analysis were performed to evaluate the correlation of MR-proADM values ​​at admission above the cut-off of 1 ng/mL in patients with localized infection and above the cut-off of 2 ng/mL and 3 ng/mL in septic patients, because they are those to which the best sensitivity, specificity and like-hood ratio values ​​correspond, chosen on the basis of recent meta-analyses and also our previous studies”.

Li P, Wang C, Pang S. The diagnostic accuracy of mid-regional pro-adrenomedullin for sepsis: a system­atic review and meta-analysis. Minerva Anestesiol 2021;87:1117-27. DOI: 10.23736/S0375-9393.21.15585-3)

Baldirà, J.; Ruiz-Rodríguez, J.C.; Ruiz-Sanmartin, A.; Chiscano, L.;Cortes, A.; Sistac, D.Á.; Ferrer-Costa,R.; Comas, I.; Villena, Y.; Larrosa,M.N.; et al. Use of Biomarkers to Improve 28-Day Mortality Stratification in Patients with Sepsis and SOFA _ 6. Biomedicines 2023, 11,2149. https://doi.org/10.3390/ biomedicines11082149

Angeletti S, Dicuonzo G, Fioravanti M, De Cesaris M, Fogolari M, Lo Presti A, et al. Procalcitonin, MR-Proadre­nomedullin, and Cytokines Measurement in Sepsis Diag­nosis: Advantages from Test Combination. Dis Markers 2015;2015:951532. 

Sargentini V, Collepardo D, D Alessandro M, Petralito G, Ceccarelli G, Alessandri F, et al. Role of biomarkers in adult sepsis and their application for a good laboratory practice: a pilot study. J Biol Regul Homeost Agents 2017;31:1147–54.

Spoto S, Nobile E, Carnà EPR, Fogolari M, Caputo D, De Florio L, Valeriani E, Benvenuto D, Costantino S, Ciccozzi M, Angeletti S. Best diagnostic accuracy of sepsis combining SIRS criteria or qSOFA score with Procalcitonin and Mid-Regional pro-Adrenomedullin outside ICU. Sci Rep. 2020 Oct 6;10(1):16605. doi: 10.1038/s41598-020-73676-y. PMID: 33024218; PMCID: PMC7538435.

Valenzuela-Sánchez F, Valenzuela-Méndez B, Bohollo de Austria R, Rodríguez-Gutiérrez JF, Estella-García Á, Fernández-Ruiz L, González-García MÁ, Rello J. Plasma levels of mid-regional pro-adrenomedullin in sepsis are associated with risk of death. Minerva Anestesiol. 2019 Apr;85(4):366-375. doi: 10.23736/S0375-9393.18.12687-3. Epub 2018 Sep 10. PMID: 30207133.)

All continuous laboratory and clinical variables (of septic patients with or without septic shock, and of septic patients vs control patients) were compared using the non-parametric Mann-Whitney test  and the results were represented as median and interquartile range (i.e. 25th-75th percentile, IQR). Categorical variables are reported as counts and percentages and assessed by chi-square or Fisher exact tests. P value <0.05 were considered as significant.

Area under the receiver operating characteristic (AUROC) curves were used to identify the biomarker or clinical score with the greatest predictive value for each endpoint, with 95% confidence intervals (95% CI) compared to determine significance. Youden’s criterion established optimal cut-of values with corresponding sensitivity and specificity values.

An AUROC of 0.5 was considered non-predictive, and 1.0 was considered a perfect predictive ability. An AUROC of 0.70 to 0.80 was considered acceptable.

Receiver operating characteristic (ROC) analysis was performed among independent variables associated with organ damage and mortality to define the optimal cut-off point for plasma MR-proADM.

-The authors acknowledged the lack of validation of their findings using an independent cohort.  This represents a major limitation of the study and raises concerns about the clinical usefulness of the data obtained.

 Reply: We thank the reviewer for this important limitation that we have added in the "limitations" section and which we hope can be filled by further future studies.We have added within the “Limitations and perspectives”:“The main limitation of our study is the absence of an external validation cohort. To ascertain the generalizability of the cut off we identified within our population, future investigations will be necessary, encompassing diverse centers and settings.”

-It is unclear why repeated measurements of MR-proADM cannot be performed. Would MR-proADM levels reflects changes in clinical symptoms?

 Reply: We thank the reviewer for allowing us this clarification. Recent studies highlight a prognostic correlation on the decline or otherwise of MR-proADM based on the clinical response to treatment for sepsis. We chose not to carry out a serial dosage of MR-proADM because it was outside the aim of the study and on the basis of the following literature data. We have added this data in the materials and methods.

 “MR-proADM measurement was performed only at admission (T=0), since the marker has a slow clearance (the stability of MR-proADM is at least 75 days in the absence of clinical changes). Therefore, it can only be measured once at the time of patient hospitalization”. [Morgenthaler, N.G.; Struck, J.; Alonso, C.; Bergmann, A. Measurement of Midregional Proadrenomedullin in Plasma with an Immunoluminometric Assay. Clin. Chem. 2005, 51, 1823–1829].

-What would be the outcome if the authors applied the cut-off value (>1.4 ng/ml) used for the non-septic group to the sepsis or septic shock groups?

Reply: The cut-off of MR-proADM >1.4 ng/ml is lower in non-septic group because these patients are less critics than septic ones. In fact, MR-proADM cut-off in septic and septic shock patients is significantly higher. This higher cut-off is confirmed also in previous studies.

Li P, Wang C, Pang S. The diagnostic accuracy of mid-regional pro-adrenomedullin for sepsis: a system­atic review and meta-analysis. Minerva Anestesiol 2021;87:1117-27. DOI: 10.23736/S0375-9393.21.15585-3)

 The use of the value of >1.4 ng/mL in the septic population would lead to the identification of false positives of sepsis, therefore it was not carried out.

However, it is the univariate and multivariate analysis that identifies the specific MR-proADM cut-off.

- In spite of the authors` statement that “the discussion was correct”, the Discussion lacks critical comparison of the findings with those in the published literature. For instance, how do the cut-off values compare and which one could then be used to guide diagnosis/prognosis?

Reply: We thank the reviewer for allowing us to expand on this crucial point.We add as follows:

-“ The use of biomarkers assists in the clinical diagnosis of infection. Clinical sepsis scores allow septic patients to be stratified based on the number and severity of organ damage, therefore they become positive later than the elevation of MR-proADM which, moreover, can be affected by oxidative stress due to other even non-infectious causes.

In our study the established cut off value of MR-proADM (with corresponding the best sensitivity and specificity value) to identify patients with localized infection was ≥ 1.44 ng/ml, to identify patients with sepsis or organ failure or not survived at 90-day was ≥ 2 ng/ml.

The use of these cut-offs could indicate timely and intensive treatment, avoiding the onset of sepsis and organ damage and/or death. These results are in line with a recent meta-analysis and systematic review evaluated the diagnostic value of MR-proADM in sepsis, finding that MR-proADM is an excellent biomarker for the diagnosis of sepsis.

Li P, Wang C, Pang S. The diagnostic accuracy of mid-regional pro-adrenomedullin for sepsis: a system­atic review and meta-analysis. Minerva Anestesiol 2021;87:1117-27. DOI: 10.23736/S0375-9393.21.15585-3)

Baldirà, J.; Ruiz-Rodríguez, J.C.; Ruiz-Sanmartin, A.; Chiscano, L.;Cortes, A.; Sistac, D.Á.; Ferrer-Costa,R.; Comas, I.; Villena, Y.; Larrosa,M.N.; et al. Use of Biomarkers to Improve 28-Day Mortality Stratification in Patients with Sepsis and SOFA _ 6. Biomedicines 2023, 11,2149. https://doi.org/10.3390/ biomedicines11082149

Angeletti S, Dicuonzo G, Fioravanti M, De Cesaris M, Fogolari M, Lo Presti A, et al. Procalcitonin, MR-Proadre­nomedullin, and Cytokines Measurement in Sepsis Diag­nosis: Advantages from Test Combination. Dis Markers 2015;2015:951532. 

Sargentini V, Collepardo D, D Alessandro M, Petralito G, Ceccarelli G, Alessandri F, et al. Role of biomarkers in adult sepsis and their application for a good laboratory practice: a pilot study. J Biol Regul Homeost Agents 2017;31:1147–54.

Spoto S, Nobile E, Carnà EPR, Fogolari M, Caputo D, De Florio L, Valeriani E, Benvenuto D, Costantino S, Ciccozzi M, Angeletti S. Best diagnostic accuracy of sepsis combining SIRS criteria or qSOFA score with Procalcitonin and Mid-Regional pro-Adrenomedullin outside ICU. Sci Rep. 2020 Oct 6;10(1):16605. doi: 10.1038/s41598-020-73676-y. PMID: 33024218; PMCID: PMC7538435.

Valenzuela-Sánchez F, Valenzuela-Méndez B, Bohollo de Austria R, Rodríguez-Gutiérrez JF, Estella-García Á, Fernández-Ruiz L, González-García MÁ, Rello J. Plasma levels of mid-regional pro-adrenomedullin in sepsis are associated with risk of death. Minerva Anestesiol. 2019 Apr;85(4):366-375. doi: 10.23736/S0375-9393.18.12687-3. Epub 2018 Sep 10. PMID: 30207133.

In the sepsis population, using 1-1.5 nmol/L as the cut-off value of MR-proADM had a higher combined sensitivity and specificity for the diagnosis of sepsis, which were 0.83 and 0.90, respectively. Li P, Wang C, Pang S. The diagnostic accuracy of mid-regional pro-adrenomedullin for sepsis: a system­atic review and meta-analysis. Minerva Anestesiol 2021;87:1117-27. DOI: 10.23736/S0375-9393.21.15585-3)

Worthy of note is the essential aid that can provide use of a cut off of MR-proADM ≥ 1.5 nmol/L for early sepsis diagnosis in those with a negative SOFA score. In this study, indeed, about 35% of patients were negative for SIRS criteria or q-SOFA, and SOFA score or for all, despite evidence of positive blood culture and documented microbiological isolate or clinical diagnosis of infection. In these patients, the use of MR-proADM was crucial to provide early diagnosis and confirm the suspicion of sepsis. Spoto S, Nobile E, Carnà EPR, Fogolari M, Caputo D, De Florio L, Valeriani E, Benvenuto D, Costantino S, Ciccozzi M, Angeletti S. Best diagnostic accuracy of sepsis combining SIRS criteria or qSOFA score with Procalcitonin and Mid-Regional pro-Adrenomedullin outside ICU. Sci Rep. 2020 Oct 6;10(1):16605. doi: 10.1038/s41598-020-73676-y. PMID: 33024218; PMCID: PMC7538435.

Spoto, S.; Fogolari, M.; De Florio, L.; Minieri, M.; Vicino, G.; Legramante, J.; Lia, M.S.; Terrinoni, A.; Caputo, D.; Costantino, S.; et al. Procalcitonin and MR-proAdrenomedullin Combination in the Etiological Diagnosis and Prognosis of Sepsis and Septic Shock. Microbial Pathogenesis 2019, 137, 103763, doi:10.1016/j.micpath.2019.103763.

MR-proADM had a high accuracy in identifying both 28-day and 90-day mortality, compared to all other biomarkers and clinical scores.

Baldirà, J.; Ruiz-Rodríguez,J.C.; Ruiz-Sanmartin, A.; Chiscano, L.;Cortes, A.; Sistac, D.Á.; Ferrer-Costa,R.; Comas, I.; Villena, Y.; Larrosa, M.N.; et al. Use of Biomarkers to Improve 28-Day Mortality Stratification in Patients with Sepsis and SOFA _ 6. Biomedicines 2023, 11,

  1. https://doi.org/10.3390/biomedicines11082149

Valenzuela-Sánchez F, Valenzuela-Méndez B, Bohollo de Austria R, Rodríguez-Gutiérrez JF, Estella-García Á, Fernández-Ruiz L, González-García MÁ, Rello J. Plasma levels of mid-regional pro-adrenomedullin in sepsis are associated with risk of death. Minerva Anestesiol. 2019 Apr;85(4):366-375. doi: 10.23736/S0375-9393.18.12687-3. Epub 2018 Sep 10. PMID: 30207133.

Outside ICU, a MR-proADM cut-off values > 3.39 nmol/L in sepsis and > 4.33 nmol/L in septic shock were associated with significant higher risk of 90-days mortality. Spoto, S.; Fogolari, M.; De Florio, L.; Minieri, M.; Vicino, G.; Legramante, J.; Lia, M.S.; Terrinoni, A.; Caputo, D.; Costantino, S.; et al. Procalcitonin and MR-proAdrenomedullin Combination in the Etiological Diagnosis and Prognosis of Sepsis and Septic Shock. Microbial Pathogenesis 2019, 137, 103763, doi:10.1016/j.micpath.2019.103763.

In ICU patients, admitted with SIRS and organ dysfunction, an MR-proADM cut-off point of 1.425 nmol/L helps to identify those with sepsis, while an MR-proADM value above 5.626 nmol/L, 48 hours after admission, was associated with a high risk of death.

Valenzuela-Sánchez F, Valenzuela-Méndez B, Bohollo de Austria R, Rodríguez-Gutiérrez JF, Estella-García Á, Fernández-Ruiz L, González-García MÁ, Rello J. Plasma levels of mid-regional pro-adrenomedullin in sepsis are associated with risk of death. Minerva Anestesiol. 2019 Apr;85(4):366-375. doi: 10.23736/S0375-9393.18.12687-3. Epub 2018 Sep 10. PMID: 30207133.

In our study the established cut off value of MR-proADM (with corresponding the best sensitivity and specificity value) to identify patients with localized infection was ≥ 1.44 ng/ml, to identify patients with sepsis with SOFA score ≥ 2, with need for ICU transfer or not survived at 90-day was ≥ 2 ng/ml”.

- Limitations of control group (beyond age and sex) should also be discussed.

Reply: We thank the reviewer for this careful analysis.Control group was chosen as real-life patients matched for age sex as much as possible. For example age comparison in patients with septic shock and control group is not significant (p=0.89).We have added this data in the "limits".

-The text will benefit from editing for English.

Reply:  English language has been further improved and corrected.

Round 3

Reviewer 1 Report

Comments and Suggestions for Authors

no further comments

Comments on the Quality of English Language

can be improved

Author Response

We thank the Reviewer for his patience and in-depth study of our work.

We believe in his suggestions which we appreciated and included in the manuscript in the Discussion section.

We have no other way of being able to modify the statistical analysis that express the validity of our results, also in line with the limited literature of this new and promising biomarker.

We will take the Reviewer's valuable suggestions into consideration for further studies.

We hope that the Reviewer also appreciates, like us, the usefulness and importance of our scientific contribution.

We have added the following sentences to the study: “These MR-proADM cut offs also correspond to those found in another recent work by S. Graziadio in which a MR-proADM value ​​greater than 1.5 nmol/L correlates with acuity increase, while greater than 1.89 nmol/L correlates with deterioration event in patients admitted to hospital with a mild to moderately severe acute illness corresponding to a National Early Warning Score (NEWS) between 2 and 5 [73].“ Furthermore, our study identifies the specific MR-proADM cut-off for each organ damage in septic patients, compared to Graziadio's study in which the MR-proADM cut-off for deterioration is evaluated in all patients admitted to hospital with a mild to moderately severe acute illness (even not infected), that is, with National Early Warning Score (NEWS) between 2 and 5.

We also attach the revised manuscript with the corrections highlighted in green and the point by point corrections inserted in the revised manuscript in the comments.

Corrections point by point

-m.francesconi@policlinicocampus.it (M.Fr.) to differentiate from Marta Fogolari (M.F).

The order of authors are different from the one submitted online at susy.mdpi.com. Please confirm which one is correct. The order of the authors on the manuscript is the correct one. Here is the correct one:

  Stefania Basili 2,†, Roberto Cangemi 2,†, Giorgio D’Avanzo 1, Domenica Marika Lupoi 1, Giulio Francesco Romiti 2, Josepmaria Argemi 3, José Ramón Yuste Ara 4,5, Felipe Lucena 3, Luciana Locorriere 1, Francesco Masini 1, Giulia Testorio 1, Rodolfo Calarco 1, Marta Fogolari 6,7, Maria Francesconi 6,7, Giulia Battifoglia 1, Sebastiano Costantino 1 and Silvia Angeletti 6,7

We confirm the email addresses.Please add the postal code (or ZIP code in the U.S.). If the postal code is not available, Post Office Box number can be added instead. We have added the postal code.

m.francesconi@policlinicocampus.it (M.Fr.) to differentiate from Marta Fogolari (M.F).

Only one paragraph is allowed in the abstract, please confirm. The abstract consists of 9 paragraphs.Abstract is attached:Please confirm if this should be em-dash, please revise them in the paper. We have changed the abstract and the paper with brackets (ICU) instead of dashes.Keywords: we have added the following keywordswe moved the results before the materials and methods section.We confirm that the numbering of figures and tables corresponds.Fig.2A….Please confirm the suggested revision. We confirm that the figures and tables are correctly numbered.Fig.2. Please change P to lowercase italicized. It is not possible to change the P to p as would be correct because the statistical analysis system automatically imposes the capitalization of p.There is no (D) explanation, please add them. We have added panel D.Please confirm if the explanation of the dotted line/colors needs to be added in the figure caption. No, no further explanation is needed.It is not possible to change the P to p as would be correct because the statistical analysis system automatically imposes the capitalization of p.

We added panel D explanation.

No, no further explanation is needed.

Please confirm which figure should be mentioned here. We added Fig. 3CFig.3. Please change P to lowercase italicized. It is not possible to change the P to p as would be correct because the statistical analysis system automatically imposes the capitalization of p.

Table 2. Please add an explanation for ns in the table footer.

We have added explanation about ns. *ns: not significant.

Materiali e metodi: Section headings should be numbered sequentially, e.g., Section 2.1, Section 2.2.1. Please confirm this revision. We have renumbered the sections corresponding to the shift of the “results”.Eessed on: Please provide the access date of the URL in the following format: “URL (accessed on Day Month Year)”. We have added the requested information to the version in use.References: References 68–72 have no citation, please add them in order. We added from 65 to 73.There is no Supplementary Materials citation in the paper, please revised. We have removed this paragraph.

We have corrected and checked the following sections: Author Contributions

Author Contributions:

Author Contributions

Conceptualization, Spoto Silvia and Silvia Angeletti; Data curation, Spoto Silvia, Stefania Basili, Roberto Cangemi, Giorgio D’Avanzo, Domenica Lupoi and Silvia Angeletti; Formal analysis, Stefania Basili, Roberto Cangemi and Silvia Angeletti; Investigation, Luciana Locorriere, Francesco Masini, Giulia Testorio, Rodolfo Calarco Giulia Battifoglia and Sebastiano Costantino; Methodology, Spoto Silvia, Stefania Basili, Roberto Cangemi and Silvia Angeletti; Project administration, Spoto Silvia and Silvia Angeletti; Software, Giorgio D’Avanzo, Domenica Lupoi, Giulio Francesco Romiti, Marta Fogolari and Maria Francesconi; Supervision, Spoto Silvia, Stefania Basili, Roberto Cangemi and Silvia Angeletti; Validation, Spoto Silvia, Stefania Basili, Roberto Cangemi and Silvia Angeletti; Visualization, Spoto Silvia, Stefania Basili, Roberto Cangemi, Josepmaria Argemi, José Ramón Yuste Ara, Juan Felipe Lucena and Silvia Angeletti; Writing – original draft, Spoto Silvia and Silvia Angeletti; Writing – review & editing, Spoto Silvia, Stefania Basili, Roberto Cangemi, José Ramón Yuste Ara and Silvia Angeletti.

We have corrected and checked the following sections: Funding

Funding

This research was funded by the Ministry of Education, Universities and Research - PRIN- 2017ATZ2YK. We acknowledge co-funding from Next Generation EU in the context of the National Recovery and Resilience Plan, Investment PE8—Project Age-It: “Ageing Well in an Ageing Society”. This resource was co-financed by the Next Generation EU (DM 1557 11.10.2022). The views and opinions expressed are only those of the authors and do not necessarily reflect those of the European Union or the European Commission. Neither the European Union nor the European Commission can be held responsible for them.

We have corrected and checked the following sections: Institutional Review Board Statement

Institutional Review Board Statement

The study was conducted according to the guidelines of the Declaration of Helsinki and was approved by the by the Local Ethics Committee of the Fondazione Policlinico Universitario Campus Bio-Medico (N° 23.17 TS).

We have corrected and checked the following sections: Informed Consent Statement

Informed Consent Statement

Written informed consent has been obtained from the patients to publish this paper.

We have corrected and checked the following sections: Data Availability Statement

Data Availability Statement

The data are contained within the article. The further research data presented in this study are available upon request from the corresponding author.

We have corrected and checked the following sections: Conflicts of Interest

Conflicts of Interest

The authors declare no conflict of interest.

Acknowledgments

We thank Stefano Spoto and Francesco Masini for English language revisions.

Abbreviations

Acute heart failure (AHF); areas under the curve (AUC); C-reactive protein (CRP); length of stay (LOS); left ventricular ejection fraction (LVEF); mid-regional pro-adrenomedullin (MR-proADM); receiver operating characteristic (ROC).

   References. We have corrected the references. the part in red must be eliminated.We confirm changes. we have also added references 73.

Reviewer 3 Report

Comments and Suggestions for Authors

In spite of the revision, major limitations remain unaddressed. In particular, proper biomarker research requires two cohorts, one to establish cut-off values
("discovery arm" and a second larger cohort to validate the findings. In the absence of this cohort, the authors` suggestions can be queried. Adding another cut-off value to those derived from previous meta-analyses does not seem to facilitate diagnosis of sepsis or predicting outcome.

Another concern is the multiple cut-off values for different parameters. It is difficult to imagine how clinical assessment could depend on sliding scale values. Once again, validating these cut-off values in an independent cohort is required.

  •  

Comments on the Quality of English Language

The text will benefit from some editing.

Author Response

We thank the Reviewer for his patience and in-depth study of our work.

We believe in his suggestions which we appreciated and included in the work both as an added value of the study and in the Discussion section.

We have no other way of being able to modify the statistical analysis that express the validity of our results, also in line with the limited literature of this new and promising biomarker.

We will take the Reviewer's valuable suggestions into consideration for further studies.

We hope that the Reviewer also appreciates, like us, the usefulness and importance of our scientific contribution.

We have added the following sentences to the study:

“These MR-proADM cut offs also correspond to those found in another recent work by S. Graziadio in which a MR-proADM value ​​greater than 1.5 nmol/L correlates with acuity increase, while greater than 1.89 nmol/L correlates with deterioration event in patients admitted to hospital with a mild to moderately severe acute illness corresponding to a National Early Warning Score (NEWS) between 2 and 5 [73].“ 

“The added value of our study is the effort to establish a threshold in the evaluation of MR-proADM that may allow for different management of patients with sepsis. In the clinical use, the MR-proADM value individuate septic patients at higher risk of death by identifying who may also benefit from more strictly medical treatment including hemodynamic management, infection source control and hard and timely antibiotic therapy, modulation of host response therapy also with adrecizumab, with administration as early as possible upon evidence of sepsis with AKI, impaired GCS or shock [55–58].”

We also attach the revised manuscript with the corrections highlighted in green and the point by point corrections inserted in the revised manuscript in the comments.

Corrections point by point

-m.francesconi@policlinicocampus.it (M.Fr.) to differentiate from Marta Fogolari (M.F).

The order of authors are different from the one submitted online at susy.mdpi.com. Please confirm which one is correct. The order of the authors on the manuscript is the correct one. Here is the correct one:

  Stefania Basili 2,†, Roberto Cangemi 2,†, Giorgio D’Avanzo 1, Domenica Marika Lupoi 1, Giulio Francesco Romiti 2, Josepmaria Argemi 3, José Ramón Yuste Ara 4,5, Felipe Lucena 3, Luciana Locorriere 1, Francesco Masini 1, Giulia Testorio 1, Rodolfo Calarco 1, Marta Fogolari 6,7, Maria Francesconi 6,7, Giulia Battifoglia 1, Sebastiano Costantino 1 and Silvia Angeletti 6,7

We confirm the email addresses.Please add the postal code (or ZIP code in the U.S.). If the postal code is not available, Post Office Box number can be added instead. We have added the postal code.

m.francesconi@policlinicocampus.it (M.Fr.) to differentiate from Marta Fogolari (M.F).

Only one paragraph is allowed in the abstract, please confirm. The abstract consists of 9 paragraphs.Abstract is attached:Please confirm if this should be em-dash, please revise them in the paper. We have changed the abstract and the paper with brackets (ICU) instead of dashes.Keywords: we have added the following keywordswe moved the results before the materials and methods section.We confirm that the numbering of figures and tables corresponds.Fig.2A….Please confirm the suggested revision. We confirm that the figures and tables are correctly numbered.Fig.2. Please change P to lowercase italicized. It is not possible to change the P to p as would be correct because the statistical analysis system automatically imposes the capitalization of p.There is no (D) explanation, please add them. We have added panel D.Please confirm if the explanation of the dotted line/colors needs to be added in the figure caption. No, no further explanation is needed.It is not possible to change the P to p as would be correct because the statistical analysis system automatically imposes the capitalization of p.

We added panel D explanation.

No, no further explanation is needed.

Please confirm which figure should be mentioned here. We added Fig. 3CFig.3. Please change P to lowercase italicized. It is not possible to change the P to p as would be correct because the statistical analysis system automatically imposes the capitalization of p.

Table 2. Please add an explanation for ns in the table footer.

We have added explanation about ns. *ns: not significant.

Materiali e metodi: Section headings should be numbered sequentially, e.g., Section 2.1, Section 2.2.1. Please confirm this revision. We have renumbered the sections corresponding to the shift of the “results”.Eessed on: Please provide the access date of the URL in the following format: “URL (accessed on Day Month Year)”. We have added the requested information to the version in use.References: References 68–72 have no citation, please add them in order. We added from 65 to 73.There is no Supplementary Materials citation in the paper, please revised. We have removed this paragraph.

We have corrected and checked the following sections: Author Contributions

Author Contributions:

Author Contributions

Conceptualization, Spoto Silvia and Silvia Angeletti; Data curation, Spoto Silvia, Stefania Basili, Roberto Cangemi, Giorgio D’Avanzo, Domenica Lupoi and Silvia Angeletti; Formal analysis, Stefania Basili, Roberto Cangemi and Silvia Angeletti; Investigation, Luciana Locorriere, Francesco Masini, Giulia Testorio, Rodolfo Calarco Giulia Battifoglia and Sebastiano Costantino; Methodology, Spoto Silvia, Stefania Basili, Roberto Cangemi and Silvia Angeletti; Project administration, Spoto Silvia and Silvia Angeletti; Software, Giorgio D’Avanzo, Domenica Lupoi, Giulio Francesco Romiti, Marta Fogolari and Maria Francesconi; Supervision, Spoto Silvia, Stefania Basili, Roberto Cangemi and Silvia Angeletti; Validation, Spoto Silvia, Stefania Basili, Roberto Cangemi and Silvia Angeletti; Visualization, Spoto Silvia, Stefania Basili, Roberto Cangemi, Josepmaria Argemi, José Ramón Yuste Ara, Juan Felipe Lucena and Silvia Angeletti; Writing – original draft, Spoto Silvia and Silvia Angeletti; Writing – review & editing, Spoto Silvia, Stefania Basili, Roberto Cangemi, José Ramón Yuste Ara and Silvia Angeletti.

We have corrected and checked the following sections: Funding

Funding

This research was funded by the Ministry of Education, Universities and Research - PRIN- 2017ATZ2YK. We acknowledge co-funding from Next Generation EU in the context of the National Recovery and Resilience Plan, Investment PE8—Project Age-It: “Ageing Well in an Ageing Society”. This resource was co-financed by the Next Generation EU (DM 1557 11.10.2022). The views and opinions expressed are only those of the authors and do not necessarily reflect those of the European Union or the European Commission. Neither the European Union nor the European Commission can be held responsible for them.

We have corrected and checked the following sections: Institutional Review Board Statement

Institutional Review Board Statement

The study was conducted according to the guidelines of the Declaration of Helsinki and was approved by the by the Local Ethics Committee of the Fondazione Policlinico Universitario Campus Bio-Medico (N° 23.17 TS).

We have corrected and checked the following sections: Informed Consent Statement

Informed Consent Statement

Written informed consent has been obtained from the patients to publish this paper.

We have corrected and checked the following sections: Data Availability Statement

Data Availability Statement

The data are contained within the article. The further research data presented in this study are available upon request from the corresponding author.

We have corrected and checked the following sections: Conflicts of Interest

Conflicts of Interest

The authors declare no conflict of interest.

Acknowledgments

We thank Stefano Spoto and Francesco Masini for English language revisions.

Abbreviations

Acute heart failure (AHF); areas under the curve (AUC); C-reactive protein (CRP); length of stay (LOS); left ventricular ejection fraction (LVEF); mid-regional pro-adrenomedullin (MR-proADM); receiver operating characteristic (ROC).

   References. We have corrected the references. the part in red must be eliminated.We confirm changes. we have also added references 73.